# OFFLINE ADAPTIVE POLICY LEANING IN REAL-WORLD SEQUENTIAL RECOMMENDATION SYSTEMS

## ABSTRACT

The training process of RL requires many trial-and-errors that are costly in real-world applications. To avoid the cost, a promising solution is to learn the policy from an offline dataset, e.g., to learn a simulator from the dataset, and train optimal policies in the simulator. By this approach, the quality of policies highly relies on the fidelity of the simulator. Unfortunately, due to the stochasticity and unsteadiness of the real-world and the unavailability of online sampling, the distortion of the simulator is inevitable. In this paper, based on the model learning technique, we propose a new paradigm to learn an RL policy from offline data in the real-world sequential recommendation system (SRS). Instead of increasing the fidelity of models for policy learning, we handle the distortion issue via learning to adapt to diverse simulators generated by the offline dataset. The adaptive policy is suitable to real-world environments where dynamics are changing and have stochasticity in the offline setting. Experiments are conducted in synthetic environments and a real-world ride-hailing platform. The results show that the method overcomes the distortion problem and produces robust recommendations in the unseen real-world.

## 1 INTRODUCTION

Recent studies have shown that reinforcement learning (RL) is a promising approach for real-world applications, e.g., sequential recommendation systems (SRS) (Wang et al., 2018; Zhao et al.; 2019; Cai et al., 2017), which make multiple rounds of recommendations for customers and maximize long-term recommendation performance. However, the high trial-and-error costs in the real-world obstruct further applications of RL methods (Strehl et al., 2006; Levine et al., 2018).

Offline (batch) RL is to learn policies with a static dataset collected by behavior policies without additional interactions with the environment (Levine et al., 2020; Siegel et al.; Wang et al., 2020; Kumar et al., 2019). Since it avoids costly trial-and-errors in the real environment, offline RL algorithms are promising to cost-sensitive applications (Levine et al., 2020). One scheme of offline RL is learning a simulator from the dataset. In this way, RL policies can be learned from the simulator directly. Although prior works on model-based learning have achieved significant efficiency improvements in online RL by learning dynamics models (Kaiser et al., 2020; Wang et al., 2019; Heess et al.; Luo et al., 2019), building an accurate simulator is still difficult, especially in offline RL. In particular, the offline dataset may not cover the whole state-action space, and there is no way for sampling in the real-world to recover the prediction error of the learned simulator. The learned policies tend to exploit regions where insufficient data are available, which causes the instability of policy learning (Kurutach et al., 2018; Zhang et al., 2015; Viereck et al., 2017). By overcoming the problem, recent studies in offline model-based RL (Yu et al., 2020; Kidambi et al., 2020) have made significant progress in MuJoCo environments (Todorov et al., 2012). These methods learn policies with uncertainty penalties. The uncertainty here is a function to evaluate the confidence of prediction correctness, which often implemented with ensemble techniques (Lowrey et al., 2019; Osband et al., 2018). By giving large rewards penalty (Yu et al., 2020) or trajectory truncation (Kidambi et al., 2020) with large uncertainty on dynamics models, policy exploration is constrained in the regions where the uncertainty of model prediction is small, so that to avoid optimizing policy to exploit regions with bad generalization ability.

However, in real-world applications like SRS, several realistic problems of the current offline learning methods are ignored. First, take SRS as an example, customer behaviors (i.e., the environment)

are non-stationary and thus change across different periods and locations. Therefore, besides the prediction error induced by model approximation, the transitions in the offline dataset also can be inaccurate in the future (Krueger et al., 2019; Chen et al., 2018; Zhao et al., 2018; Thomas et al., 2017; Li & de Rijke, 2019). Second, different from traditional RL environments which are overall deterministic (Brockman et al., 2016), real-world environments often introduce stochasticity. For example, after recommending a production to a customer, it is hard to model the user's decisions without stochasticity (e.g., buying it or not), even with large enough data. Hidden confounder factors (Forney et al.; Bareinboim et al., 2015; Shang et al., 2019) obstruct the deterministic predictions. As a result, the uncertainty regions are drastically increased and thus the exploration of policy learning are obscured.

In this paper, instead of constraining policy exploration in high-confidence regions, we study to handle the offline issue by learning to adapt. We propose an adaptive policy which is trained to make optimal actions efficiently in regions with high confidence for model prediction. While in regions with large uncertainty, the policy is trained to identify the representation of each dynamics model and adapt the optimal decisions on the representation. When deploying the policy in the environment, the policy identifies the dynamics in the real-world through interaction, then adapt its behavior. The module to represent the dynamics models is named environment-context extractor. The extractor and adaptive policy should be learned in a diverse simulator set and thus can generalize to unknown situations. As a solution, we propose to use model-learning techniques with augmentation approaches to generate a simulator set to cover real-world situations. In this way, with a sufficiently large simulator set, the learned policy can adapt robustly in unknown real-world environments.

To learn the adaptive policy and environment-context extractor, we first analyze and formulate the environment context representation problem in SRS. In SRS scenarios, the recommendation platforms interact with customers. Each customer can be regarded as an environment in the view of the RL paradigm. The environments include a two-level structure: in the high-level structure, a recommendation platform serves customers from multiple domains (e.g., in different cities and countries). In the low-level structure (i.e., for each domain), there are numerous customers with different behaviors, and the behaviors are dependent on the domain they current in. Although there have been recent interests in learning the representation of environment parameters based on the agent's trajectories in the robotics domain (Peng et al., 2018; Akkaya et al., 2019; Zhu et al., 2018; Sadeghi et al.), the environment-context representation problem in SRS has never been proposed and the two-level environment structure makes the environment context agnostic based on a single customer's trajectory without considering the domain he/she current in. As a solution, we use a special network to embed the domain information and show that the additional domain information is necessary for representing the environment contexts in SRS.

As the result, we propose Offline learning with Adaptive Policy in sequential Recommendation Systems (OapRS), as a new paradigm to solve an offline problem that policies can applied to real-world applications without any additional online sample. By learning to adapt with the representation of dynamics, OapRS is suitable to real-world scenarios in which environments are non-stationary and have stochasticity. As far as we know, this is also the first study on reality-gap and the environment-context representation problem in SRS. We conduct experiments in a reproducible synthetic environment and a real-world recommendation scenario: the driver program recommendation system of a ride-hailing platform. Our empirical evaluations demonstrate that OapRS can learn reasonable environment contexts and makes robust recommendations in unseen environments.

## 2    RELATED WORK

Reinforcement learning (RL) has shown to be a promising approach for real-world sequential recommendation systems (SRS) (Wang et al., 2018; Zhao et al.; 2019; Cai et al., 2017) to make optimal recommendations with long-term performance. However, numerous online unconstrained trial-and-errors in RL training obstruct the further applications of RL in "safety critical" SRS scenario since it may result in large economic losses (Levine et al., 2020; Gilotte et al.; Theocharous et al., 2015; Thomas et al., 2017). Many studies propose to overcome the problem by offline (batch) RL (Lange et al., 2012).

Most prior works on offline RL introduce model-free algorithms. To overcome the extrapolation error, which is introduced by the mismatch between the offline dataset and true state-action occupancy (Wang et al., 2020), these methods are designed to constrain the target policy to be close to the behavior policies (Wang et al., 2020; Kumar et al., 2019; Wu et al., 2019), apply ensemble

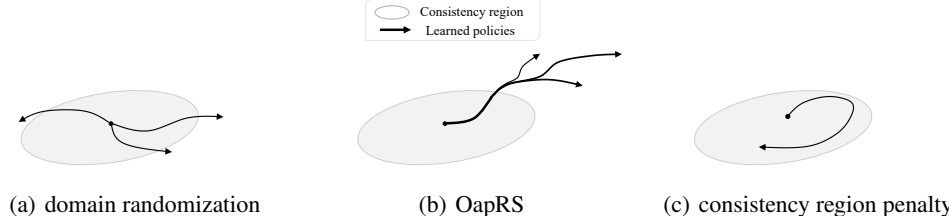

Figure 1: Illustration of the OapRS compared with other methods. The line denotes the optimal trajectory of the learned policies. There are several policies in domain randomization and OapRS since the methods learn to adapt in multiple simulators. The gray oval denotes the consistency region.

methods for robust value function estimation (Agarwal et al., 2020), or re-weight with importance sampling (Liu et al., 2019). Most recent studies have shown that learning robust policies from the approximation of dynamics models has the potential to take actions outside the distribution on behavior policies and thus can converge to better policies (Kidambi et al., 2020; Yu et al., 2020). To remedy the extrapolation error, these methods learn policies from dynamics models with uncertainty penalty. In particular, Kidambi et al. (2020) constructs terminating states based on a hard threshold on uncertainty, while Yu et al. (2020) uses a soft reward penalty to incorporate uncertainty. The uncertainty is often computed via the inconsistency of the ensemble dynamics model predictions on each state-action pair to evaluate the confidence of predictions on next states. The penalty constrains policy exploration and optimization in the regions with high consistency for better lower bound performance in the deployment environment (Kidambi et al., 2020; Yu et al., 2020). However, in SRS, the uncertainty estimation via the inconsistency will introduce two unexpected factors: (1) inconsistency of predictions may come from the non-stationary of the environment. (2) the oracle dynamics model itself may have high uncertainty (comes from the stochastic of the environment). As a result, uncertainty estimation with these compounding factors may constrain the policy to explore in a small region and thus hard to optimize.

An alternative to learning from dynamics model and deploying in real-world environments is domain randomization techniques, which is a popular solution in the robotic domain (Tobin et al., 2017; Sadeghi & Levine, 2017; Tobin et al., 2018; James et al., 2019; Peng et al., 2018; Akkaya et al., 2019; Nagabandi et al.). In general, the framework trains a policy in numerous simulators, and then deploys it to unknown target domains directly. The numerous simulators are constructed by sampling different environment parameters, which are modeled by human experts with laws of physics in the real-world. The algorithms can train a uniform policy (Tobin et al., 2017; Sadeghi & Levine, 2017; Tobin et al., 2018; James et al., 2019) to maximize the expectation of the long-term rewards of the simulators, or train an adaptive policy with online system identification (OSI) techniques (Peng et al., 2018; Akkaya et al., 2019; Yu et al.) to extract the representation of the dynamics of the simulators and maximize the performance metric of each simulator. In this paper, we adopt a similar idea of OSI to construct the environment-parameter extractor and the adaptive policy. However, in the offline setting, generating a simulator set by random sampling directly from the parameter space is not only hard since we do not have the exact knowledge of a representation of environment parameters to build the simulator, but inefficient because we only have access to the offline dataset to estimate the real-world dynamics.

The difference between the aforementioned methods and OapRS is shown in Figure 2. Compared with previous methods learned by uncertainty penalty (Kidambi et al., 2020; Yu et al., 2020) to constrain policy learning (in Figure 1(c)), we learn to adapt decisions for all possible dynamics transitions (which are learned by ensemble) in the states out of the consistency regions (in Figure 1(b)). While learning in the entire dynamics space is an ideal solution to learn a robust policy (in Figure 1(a)), for less training cost and better efficiency, the dynamics set is generated by an offline dataset and model-learning algorithms with ensemble methods. In this way, we construct a consistency region implicitly in which optimal trajectories are similar for each simulator, while the domain randomization learns to adapt in the whole space without considering the knowledge of the offline dataset.

In SRS, the non-stationarity and stochasticity problems have been studied recently. On one hand, Zhao et al. (2018); Chen et al. (2019) propose to handle the non-stationarity problem by optimizing the policy with offline data directly and correct the policy by learning with additional online feedback

data from customers. Our work focus on learning an adaptive policy from offline data directly, which is important for safety critical applications. On the other hand, the stochasticity problem can be modeled with confounder factors. Recent works show policy can make better decisions considering hidden confounder factors when optimizing policy (Shang et al., 2019; Forney et al.; Bareinboim et al., 2015). In this work, we consider stochasticity by learning dynamics models with hidden confounder factors (Shang et al., 2019).

# 3 BACKGROUND AND NOTATIONS

In the standard RL framework, an agent interacts with an environment described as a Markov Decision Process (MDP) (Sutton & Barto, 2018). The agent learns a policy $\pi(a_t|s_t)$, which chooses an action $a_t \in \mathcal{A}$ conditional on a particular state $s_t \in \mathcal{S}$, at each time-step $t \in \{0, 1, ..., T\}$. $\mathcal{S}$ and $\mathcal{A}$ denote the state and action spaces, respectively. The reward function $r_t = r(s_t, a_t) \in \mathcal{R}$ evaluates the immediate performance of action $a_t$ given state $s_t$. The goal of RL is to find an optimal policy $\pi^*$, which maximizes the multi-step cumulative discounted reward (i.e., long-term performance). The objective is to maximize $J_\rho(\pi) = \max_\pi \mathbb{E}_{\tau \sim p(\tau|\pi,\rho)} \left[ \sum_{k=0}^{T} \gamma^k r_{t+k} \right]$, where $\gamma$ denotes the discount factor. $p(\tau|\pi, \rho)$ is the probability of generating a trajectory $\tau := [s_0, a_0, ..., a_{T-1}, s_T]$ under the policy $\pi$ and a transition dynamics $\rho(s_{t+1}|s_t, a_t)$, where $T$ denotes the trajectory length. In particular, $p(\tau \mid \pi) := \rho_0(s_0) \prod_{t=0}^{T} \rho(s_{t+1} \mid s_t, a_t)\pi(s_t, a_t)$, where $\rho_0(s_0)$ is the initial state distribution. A common way to find an optimal policy $\pi^*$ is by optimizing the policy parameters with gradient ascent along $\nabla J_\rho(\pi)$. SRS problem can be naturally formulated as an MDP. In general, the recommendation system is a policy $\pi$, while the environment consists of numerous human customers. During the learning process, the policy interacts with customers to execute the recommended action $a_t$ based on the customer state $s_t$.

In the offline RL problem, we assume that we are only given a static dataset $\mathcal{D} = \{(s_i, a_i, s_{i+1})\}_{i=1}^N$ collected by some unknown policies. The reward function $r$ is given by human experts. The goal is to output a sub-optimal policy $\pi^*$ to maximize $J_\rho$ with the static dataset.

SRS problem can be naturally formulated as an MDP. In general, the recommendation system is the policy $\pi$, while customers can be regarded as the environment $\rho$. During the learning process, the policy interacts with customers to execute the recommended action $a_t$ based on the customer state $s_t$. A set of recommendation business metrics define the immediate performance $r_t$ based on customers' feedback. In this work, we consider the non-stationarity of environment w.r.t time periods, which is common in SRS applications (Thomas et al., 2017). We demonstrate the phenomenon in Figure 14. We use subscript $\rho_t$ to denote the dynamics model at time-step $t$ so that the dynamics may be different in different timesteps. We also model the stochasticity with hidden confounder (Shang et al., 2019). In particular, there is a hidden policy $\pi_t^H$ for each timestep $t$ to output confounder factors: $x_t^H \sim \pi_t^H(x^H|s_t, a_t)$. The dynamics model output the next state based on $s_t$, $a_t$ and the confounder $x_t^H$, that is $s_{t+1} = \rho_t(s_t, a_t, x_t^H)$. For the notation simplification, the following we still use $s' = \rho_t(s, a)$ to denote the dynamics model with hidden confounder factors.

# 4 OAPRS: OFFLINE LEARNING WITH ADAPTIVE POLICY IN SEQUENTIAL RECOMMENDATION SYSTEM

In the offline setting, the learned policies tend to exploit regions where the model is distorted since insufficient data are available Kurutach et al. (2018). Prior offline model-based techniques have shown great potential in learning a better policy without strict constraints in action selections (Yu et al., 2020; Kidambi et al., 2020). However, the robust offline policy learning in real-world applications is still challenging for following reasons: 1) In real-world applications like SRS, environments are non-stationary, thus, besides model learning error, the transition information of the dataset can be distorted too; 2) It is hard to predict the future states with high determinacy since there are many hidden confounders in SRS (Shang et al., 2019). Since the environment has stochasticity, learning by consistency penalties will obstruct the policy into a small region for exploration.

To output a robust policy in stochastic environments, we first study to handle the distortion problem without relying on the consistency penalty and propose the learn to adapt framwork (in section 4.1). Then we formulate and analyze the environment-context representation problem in SRS and propose an environment-context extractor and an adaptive policy optimization method (in section 4.2). Finally,

we discuss important practical details to improve the robustness of simulators and policy in offline dataset (in section 4.3).

### 4.1 OFFLINE ADAPTIVE POLICY LEARNING

In this work, we assume that given a specific reward function $r$, any dynamics of environment $\rho(s'|s,a) \in \mathcal{T}$ can be fully characterized with an environment-context vector $z \in \mathcal{Z}$ (e.g., the friction coefficient in robotics), where $\mathcal{T}$ denotes a set of dynamics and $\mathcal{Z}$ denotes the space of the context vectors. Formally, there is a mapping function $\phi : \mathcal{T} \to \mathcal{Z}$. We call $\phi$ an environment-context extractor. We define the optimal environment-context extractor $\phi^*$ the one that satisfies: $\exists \pi_{\phi^*} \in \Pi, \forall \rho \in \mathcal{T}, J_\rho(\pi_{\phi^*}) = \max_\pi J_\rho(\pi)$, where $\pi_\phi := \pi_\phi(a_t|\phi(z_t|\rho_t), s_t)$ is an adaptive policy and $\Pi$ denotes the policy class. We discuss the input of $\phi$ in the next section. Besides, we define the optimal adaptive policy $\pi_{\phi^*}^*$ the one that satisfies $\forall \rho \in \mathcal{T}, J_\rho(\pi_{\phi^*}^*) = \max_\pi J_\rho(\pi)$. With the optimal $\phi^*$ and $\pi_{\phi^*}^*$, given any $\rho$ in $\mathcal{T}$, the adaptive policy can adapt to make the best decisions via the output of environment-context $z$. To achieve this, given a dynamics model set $\mathcal{T}$, we optimize $\phi$ and $\pi_\phi$ by the following objective function:

$$\phi^*, \pi_{\phi^*}^* = \arg\max_{\phi, \pi_\phi} \mathbb{E}_{\rho \sim \mathcal{T}} \left[ J_\rho(\pi_\phi) \right], \tag{1}$$

where $\sim$ denotes a sample strategy to draw dynamics models $\rho$ from the dynamics set $\mathcal{T}$ w.r.t $P[\rho] > 0, \forall \rho \in \mathcal{T}$. The strategy we used in OapRS is uniform sampling. When deployed in the real-world $\rho^r$, OapRS infers the environment contexts by $\phi^*(\rho_t^r)$ and makes the best action by $\pi_{\phi^*}^*$. If all the variant of dynamics models in the real-world $\mathcal{T}_r$ fall into $\mathcal{T}$, we can deploy the policy robustly via the optimal $\phi^*$ and $\pi_{\phi^*}^*$.

To learn the robust $\phi^*$ and $\pi_{\phi^*}^*$ for real-world deployment, the crucial problem is to construct a set of dynamics $\{\rho\}$ to cover the real-world dynamics set $\mathcal{T}_r$. One ideal solution is to construct all possible dynamics model in $\mathcal{S} \times \mathcal{A} \times \mathcal{S}$ (or all possible context vectors in $\mathcal{Z}$ and the corresponding simulators when we have the knowledge of the environment context) and optimize $\phi$ and $\pi$ via equation 1 (Tobin et al., 2017; Sadeghi & Levine, 2017; Tobin et al., 2018; James et al., 2019; Peng et al., 2018; Akkaya et al., 2019). However, in real-world applications like SRS, we have no exact knowledge to formulate customer behavior, and thus space $\mathcal{Z}$ is unknown. Besides, since the real-world fall in a subspace of the dynamics space, constructing all possible dynamics model directly is costly and inefficient for policy learning. We propose to regard model-learning algorithms $G$ as an efficient generator to map dataset to a dynamics model where the latent environment-context is near the real-world: $G(\mathcal{D}) \to \rho$ (although does not match exactly). Then we construct a diverse dynamics set by augmenting on $G$ and $\mathcal{D}$. In particular, for augmenting on $\mathcal{D}$, we divide the offline dataset by the domain knowledge (e.g., we can split the dataset by city, country, or time period in production recommendation systems); for augmenting on $G$, we can select several model-learning algorithms or different hyper-parameters for the same algorithm (Kurutach et al., 2018) to generate simulators.

We list the OapRS framework in Algorithm 1. The framework is robust to the distortion of simulators since our goal is to generate a dynamics model in which the environment contexts are near the distribution of the real-world instead of recovering the real-world dynamics exactly. The method is also robust to stochastic environments since it learns to adapt to all possible situations in the state out of the consistency region instead of doing penalty in the state with multiple situations. (as illustrated in Figure 1(b)). Additionally, we point out that the framework is orthogonal to other techniques since the reliable model-learning methods can reduce the requirement of dynamics augmentation to cover the real-world parameter space.

### 4.2 OPTIMIZATION OF ADAPTIVE POLICY WITH ENVIRONMENT-CONTEXT EXTRACTOR

In this section, we formulate the environment-context representation problem in SRS. We first define an environment-context dependent dynamics as follows: $\rho(s_{t+1}|s_t, a_t, z_t)$, where $z$ is the environment-context vector. Before learning the representation $z$ by $\phi(z|\rho)$, the question is: What is an suitable input to $\phi$ for representation learning.

In the SRS scenario, the recommendation policy interacts with customers in multiple domains. Each domain consists of numerous customers. Taking the ride-hailing platform as an example, the platform

---

**Algorithm 1** OapRS framework

---

**Input**: $\phi_\varphi$ as environment-context extractor, parameterized by $\varphi$; Adaptive policy network $\pi_\theta$ parameterized by $\theta$; Offline expert data $\mathcal{D}$; Model learning algorithm $G$; Rollout horizon $H$;minibatch $m$ for policy training
**Process**:

    divide offline dataset into $N$ partitions $\{\bar{\mathcal{D}}\}$ and generate $M \times N$ dynamics model $\{\rho\}$ by ensemble of $M$ model learning technique $G$
    **for** 1, 2, 3, ... **do**
        random select the dynamics model $\rho^i$ in $\{\rho\}$ and sample $s_1$ from $\bar{\mathcal{D}}$ which is used to train $\rho^i$
        initialize buffer $\mathcal{D}_{rollout}$
        **for** $t$=1,2,...,$H$ **do**
            sample $z_t$ from $\phi(z_t|\rho_t^i)$ and then sample $a_t$ from $\pi(a_t|s_t, z_t)$
            rollout one step $s_{t+1}, r_{t+1} \sim \rho_t^i(s_t, a_t)$ and then add $(s_{t+1}, r_{t+1}, s_t, a_t, z_t)$ to $\mathcal{D}_{rollout}$
            **if** $t \mod m == 0$ **then**
                Update $\phi$ and $\pi$ with equation 1 by policy gradient (e.g., PPO) and then $\mathcal{D}_{rollout} \leftarrow \emptyset$
            **end if**
        **end for**
    **end for**

---

provides services in multiple cities (i.e., domains), and interacts with numerous drivers in each city (i.e., customers). To train a platform recommendation policy for customers, we regard each customer as an environment $\rho$. Different from traditional agent controlling, in SRS, customer's behavior is not only dependent on the character, but the domain he/she belongs to. For instance, passengers' demand will be very different with time and cities geographically. Drivers in different cities might have different engagement (i.e., online time) due to different demand and supply, which is independent of their personas. Drivers' behavior will also change when the domain he/she belongs to changes. Based on the analysis, we propose that the input of $\phi$ should include personas information besides domain information.

In the robotics domain, environment representation approaches, called system identification, have been proposed recently (Peng et al., 2018; Akkaya et al., 2019; Yu et al.). The policy incorporates an online system identification module $\phi(z_t|s_t, \tau_{0:t})$, which utilizes a history of past states and actions $\tau_{0:t} = [s_0, a_0, ..., s_{t-1}, a_{t-1}, s_t]$ to predict the parameters of the dynamics. Since $\rho$ in SRS is also dependent on domain information, we follow the system identification technique and add the domain trajectory information $\mathcal{T}_{0:t} = [S_0, A_0, S_1, A_1, ..., S_t]$ to the input. That is: $z_t = \phi(s_t, a_{t-1}, S_t, A_{t-1}, z_{t-1})$, where $(S_t^d, A_{t-1}^d) := \{(s_t^{i,d}, a_{t-1}^{i,d})\}_{i=1}^N$ , which includes $N$ state-action pairs at each time-step $t$. We use superscript $d$ to denote the domain type.

Since the number of customers $N$ can be large and vary from timestep to timestep, it is impractical to feed $(S_t^d, A_{t-1}^d)$ to the neural network directly. In this work, we propose State-Action Distributional variational AutoEncoder (SA-DAE) to solve this problem by inferring the latent embedding $\upsilon$ of $X$. In the rest of this article, we use $X_t^d := (S_t^d, A_{t-1}^d)$ to for brevity. We first design the data generative process based on the encoded variable $\upsilon$ based on the two assumptions: First, we assume state-action pairs in $X_t^d$ are i.i.d. sampled from a distribution $p_{\psi_t^d}(s, a)$, which is parameterized by $\psi_t^d$ for each time-step $t$ and domain $d$. Second, we also assume that the parameters $\psi$ of the distribution are generated by a distribution $p_\theta(\psi|\upsilon)$, parameterized by $\theta$. It involves a latent continuous random variable $\upsilon$, which is generated from another prior distribution $p(\upsilon)$. Then the generation of $X$ includes three steps: (1) sample $\upsilon$ from $p(\upsilon)$; (2) sample $\psi$ from distribution $p_\theta(\psi|\upsilon)$ ; (3) sample $p_\psi(s, a)$ repeatedly to generate $X$.

Our target is to learn an embedding model $q_\kappa(\upsilon|X)$ parameterized by $\kappa$, aligned with the posterior approximation $p_\theta(\upsilon|X)$. Using Kullback-Leibler Divergence (KLD) as the measurement, the objective can be written as follows:

$$\min L_{\textbf{dae}} = \min \mathbb{E}_{X \sim \mathcal{D}_X} \left[ KLD \left( q_\kappa(\upsilon|X) || p_\theta(\upsilon|X) \right) \right], \tag{2}$$

where the dataset $\mathcal{D}_X := \{X_t^d : d \in \mathcal{G}, 0 < t \leq T\}$ includes state-action pairs in all time-steps $0 < t \leq T$ and domains $\mathcal{G}$, and the posterior $p_\theta(\upsilon|X)$ is the target distribution of $q_\kappa(\upsilon|X)$. Under the assumption of i.i.d. sampling on $X$, the probabilities of $q_\kappa(\upsilon|X)$ and $p_\theta(X|\upsilon)$ can be estimated

via likelihood:

$$q_\kappa(v|X) = \prod_{i=1}^{N} q_\kappa \left( v|s^{(i)}, a^{(i)} \right) \tag{3}$$

$$p_\theta(X|v) = \prod_{i=1}^{N} p_\theta \left( s^{(i)}, a^{(i)}|v \right) = \prod_{i=1}^{N} p_\psi \left( s^{(i)}, a^{(i)} \right) p_\theta(\psi|v), \tag{4}$$

where $\psi$ denotes the parameters of distribution $p_\psi$. We give our theorem on the tractable evidence lower bound (ELBO) in Theorem 1. We leave the proofs in the Appendix A.

**Theorem 1** *The tractable ELBO of state-action distributional variational inference is:*

$$\mathbb{E}_{X \sim \mathcal{D}_X} \left[ \mathbb{E}_{q_\kappa(v|X)} \left[ \sum_{i=1}^{N} \log p_\theta \left( s^{(i)}|v \right) + \log p_\theta \left( a^{(i)}|v, s^{(i)} \right) \right] + KLD \left( q_\kappa(v|X) || p_\theta(v) \right) \right] \tag{5}$$

Theorem 1 give us a three-step pipeline to minimize the objective of equation 2: (1) sample a batch of $X = \{(s, a)\}$ from expert dataset $\mathcal{D}_X$; (2) infer latent code $v$ via the likelihood probability of equation 3; (3) compute the reconstructed log-probability of state-action pairs based on equation 4 and KL divergence between posterior and prior of $v$, and then apply the gradient to $\kappa$ and $\theta$.

Finally, the function $q_\kappa$ embed the distribution of customer trajectory $\mathcal{T}_{0:t}$ into the latent code trajectory $v_{0:t}$. After that, the extractor $\phi$ can infer environment-context both with $v$ and $\tau$. Then the adaptive policy $\pi(a_t | s_t, z_t)$ samples an action based on $z_t$.

### 4.3 ROBUST OFFLINE POLICY AND MODEL LEARNING IN IMPLEMENTATION

The dynamics model and policy are high-dimensional and complex to learn. We introduce three important implementation techniques to handle these problems (more details can be found in the Appendix F): (1) Semi dynamics model learning: To reduce the model learning complexity, with domain expert knowledge, we learn the $\pi^h(\bar{s}|s, a)$ and use a hand-coded function $map(s'|\bar{s}, s, a)$ to construct the left of the states, instead of learning to predict $s'$ directly; (2) Confidence penalty: To reduce the cost of dynamics model construction, we send policy with a large negative reward when policy reach the unreal regions. The unreal region is judged by a expert rule. (3) Domain reward normalization: Reward normalization is important for model-free RL algorithms (Engstrom et al.). However, different simulators may have different reward scales. Instead of rescaling the reward with a uniform scale, we rescale the reward for each domain.

## 5 EXPERIMENTS

In this work, we focus on the three research questions: (1) In SRS scenarios, can the adaptive policy of OapRS make more robust decisions than other learn-to-adapt methods do? (2) By adapting to the learned representation of the dynamics, can OapRS handle the distortion of the constructed dynamics models? (3) Does non-steadiness and stochasticity in real-world applications actually affect the performance of offline learning? If so, does OapRS overcome this challenge? We conduct experiments in a synthetic environment and a real-world environment to answer these questions.

### 5.1 RESULTS IN SYNTHETIC ENVIRONMENT

To analyze the adaptability of OapRS and its robustness to distortion, we first conduct the experiments in a synthetic environment in which the environment parameters $z$ are configurable. The environment is called the long-term customer satisfaction (LTS) problem, a sequential recommendation environment comes from Google RecSim (Ie et al., 2019). Instead of constructing dynamics set by learning with augmentation (line 1 in Algorithm 1), for analyzing the results reasonably, we construct the training dynamics set by selecting $z$ directly, and control the difference of $z$ between the training set and the target domain to design different tasks. In particular, we construct the target dynamics with $z^{test}$ and select the training dynamics set by equidistant sampling parameters $z$ from the space and removing those $||z - z^{test}|| < \alpha$. The details of environment and experiment setup of the following can be found in Appendix B.1.

We first test the adaptability of OapRS in SRS. The algorithms we compared are (1) DR-UNI: The domain randomization technique to learn a uniform policy ((Tobin et al., 2017)); (2) DR-OSI: An online system identification algorithm proposed by Peng et al. (2018); (3) DIRECT: Train in the dynamics model where $z$ nearest to the target domain and deploy directly. (4) Upper Bound: A policy trained in the target domain directly. We regarded it as the upper bound performance. We construct three tasks: "LTS1", "LTS2" and "LTS3". The numbers after "LTS" denote the removing scale $\alpha$. We report our results in Figure 2. First, the results of DIRECT show that the performance degradation is severe in the tasks. Without considering the difference between training and deploying, the policy generates unpredictable behaviors. Second, all algorithms which consider learning from multiple dynamic models can improve the robustness in unknown domains. However, the algorithms that adapt by the representation of environments (OapRS and DR-OSI) reach better performance since they try to find the optimal policy in the representation of the environment instead of maximizing the expected performance in the training set. In addition, OapRS reaches the near-optimal performance and does better than DR-OSI in difficult tasks (e.g., LTS3). The results demonstrate that without considering the domain information, the representation can not be extracted well. In more difficult tasks, the limitation of the representation ability restricts the performance of the adaptive policy. We show more results about the SA-DAE and the extractor in Appendix C.

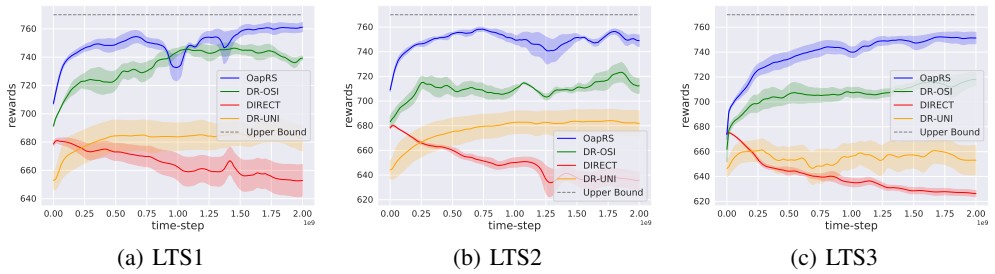

| (a) LTS1 | (b) LTS2 | (c) LTS3 |

Figure 2: Illustration of the performance in synthetic environments. The solid curves are the mean reward and the shadow is the standard error of three seeds.

We then analyze the influence of dynamics distortion on OapRS. To mimic the process of offline model-based learning, we follow the previous experiment setting at "LTS3", but divide $z$ into two part: $z_e$ and $z_d$. In particular, $z_e$ denotes the parameters that the behavior can be learned well by dynamics model, which are equidistant sampled as before. And $z_d$ denotes the distortion parameters, which are the same in all dynamic models if distortion does not exist. We generate the distorted dynamics parameter $\hat{z}_d$ by $\hat{z}_d \sim z_d + U(-\beta, \beta)$ for training, where $\beta$ can be regarded as the level of distortion. Figure 3 shows the performance of OapRS in this setting. We can see in Figure 3(a) that the deployment performance of OapRS with limited training set declines when the distortion level becomes larger, but the performance is still better than the compared methods. However, in Figure 3(b), we found that with enough sampled simulators, OapRS can overcome the distortion problem well. As a conclusion, in empirically, by augmenting enough diverse dynamics models, it is possible to overcome the distortion problem via OapRS.

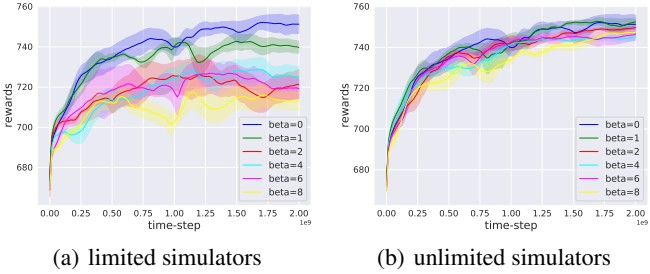

| (a) limited simulators | (b) unlimited simulators |

Figure 3: Illustration of the performance in distored synthetic environments. The solid curves are the mean reward and the shadow is the standard error of three seeds. In limited setting, we sample $\beta$ for each simulator and fix it. In unlimited setting, we sample $\beta$ for each simulator and each episode.

## 5.2 EXPERIMENTS IN A REAL-WORLD APPLICATION

We finally test OapRS in the driver program recommendation (DPR) of a large-scale ride-hailing platform in the real world (the detail of task setting can be seen Appendix B.2). The offline data comes from a human policy. We construct the dynamics model set by DEMER (Shang et al., 2019) for its ability to learn hidden confounders. For dynamics model augmentation, we divide the dataset into 34 partitions by cities and time periods. For better comparison of algorithms and demonstrate the offline learning performance in stochastic environments, we first conduct a semi-online experiment: We leave one city in the dynamics set out for policy training, and test the policy performance in the unseen city. We select three different cities for these experiments. The algorithms we compare are: (1) MOPO: an offline model-based algorithm by consistency penalties. The dynamics models are also trained in the same way as OapRS (Yu et al., 2020); (2) BCQ: an offline model-free algorithm by constraining actions (Wang et al., 2020); (3) DEMER-policy: train a policy in one single dynamics model without additional constrains but select the part of dataset nearest to the target environments to learn the dynamics model. Here we select the dataset collected in the deployment city and the nearest time period to train the dynamics model (note: the dataset are unseen to other algorithms). It is the policy learning method in the original paper of DEMER (Shang et al., 2019). Figure 4(a), Figure 4(b) and Figure 4(c) show the results. We finally deploy the policy trained by MOPO and OapRS to the real-world and test the performance for 7 days. The results is shown in Figure4(d). We deploy the policy from day 23 to day 29. Before deployment, drivers are recommended with the same human expert policy. We found that the performance improvement of the MOPO policy is 0.1%, which is similar to the expert policy, while the improvement of OapRS is 6.9%, which is significantly better than the expert. The details of the results can be seen in Table 4.

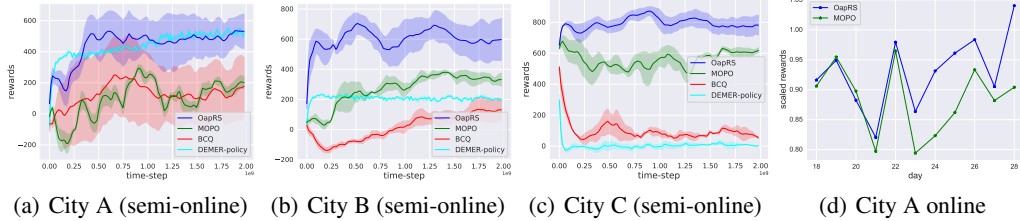

(a) City A (semi-online)   (b) City B (semi-online)   (c) City C (semi-online)   (d) City A online

Figure 4: Illustration of the performance in semi-online and online tests. In the semi-online setting, the Y-axis is the averaged long-term reward among the users in the environments. In the online test, the Y-axis is the averaged daily reward. The rewards of online data are rescaled.

## 6 DISCUSSION

In this paper, we propose a new paradigm of model-based learning to solve the offline RL in real-world applications. We focus on two realistic problems in real-world SRS applications: (1) real-world environments are non-stationary; (2) real-world environments are often with stochasticity. Instead of learning by penalty, constraint or randomization, OapRS solves the problem by learning to adapt to diverse simulators generated by the offline dataset. We formulate and analyze the environment-parameter representation learning problem in SRS and optimize the adaptive policy with the learned environment representation. The experiment results show the adaptability of OapRS, the performance decline of offline RL algorithms faced with the two realistic problems, and how OapRS work well in this situation. Although we focus on the real-world application of SRS, we consider that OapRS can be a general paradigm to solve the offline RL problem in other complex environments.

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

## A    PROOF OF SA-DAE

We design the data generative process based on the encoded variable $\upsilon$ with two assumptions: First, we assume that state-action pairs in $X$ are i.i.d. sampled from a distribution $p_\psi(s, a)$, which is parameterized by $\psi$. Second, we also assume that the distribution parameters $\psi$ are generated by a distribution $p_\theta(\psi|\upsilon)$, parameterized by $\theta$. It involves a latent continuous random variable $\upsilon$, which is generated from another prior distribution $p(\upsilon)$. Then, the generation of $X$ includes three steps: 1) sample $\upsilon$ from $p(\upsilon)$; 2) sample $\psi$ from distribution $p_\theta(\psi|\upsilon)$ ; 3) sample $p_\psi(s, a)$ repeatedly to generate $X$. A comparison with VAE on directed graphical model is shown in Figure 5.

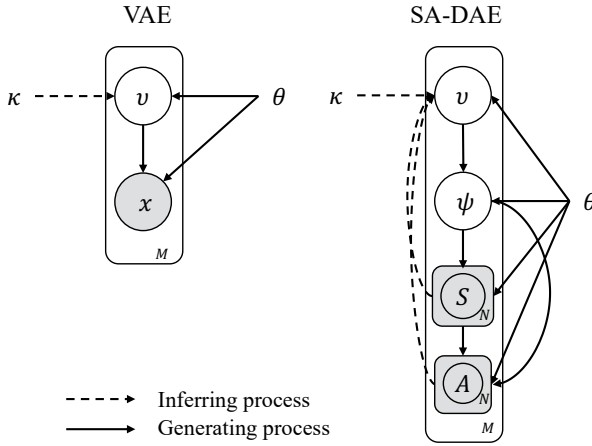

Figure 5: Comparison of SA-DAE and vanilla VAE through the directed graphical model. The circles denote the variable nodes. The rounded rectangle denotes the dataset nodes, in which the notation in the corner denotes the number of datasets. $\theta$ denotes the approximation parameters of the generative model, and $\kappa$ denotes the parameters of the variational approximation model.

Our target is to learn an embedding model $q_\kappa(\upsilon|X)$ parameterized by $\kappa$, which aligned with the posterior $p_\theta(\upsilon|X)$. Using KL divergence (KLD) as the measure of distribution, the objective function can be written as follows:

$$\min L_{\mathbf{sad}} = \min \mathbb{E}_{X \sim \mathcal{D}_X} \left[ KLD \left( q_\kappa(\upsilon|X) || p_\theta(\upsilon|X) \right) \right]. \tag{6}$$

where the dataset $\mathcal{D}_X := \{ X_t^d : d \in \mathcal{G}, 0 < t \leq T \}$ includes state-action pairs in all time-steps $0 < t \leq T$ and domains $\mathcal{G}$, and the posterior $p_\theta(\upsilon|X)$ is the target distribution of $q_\kappa(\upsilon|X)$. We provide the evidence lower bound (ELBO) of Equation equation 6 in Lemma 1.

**Lemma 1** *The ELBO of state-action distributional variational inference is:*

$$ELBO = \mathbb{E}_{X \sim \mathcal{D}_X} \left[ \mathbb{E}_{q_\kappa(\upsilon|X)} \left[ \log p_\theta(X|\upsilon) \right] - KLD \left( q_\kappa(\upsilon|X) || p_\theta(\upsilon) \right) \right]$$

**Proof A.1**

$$KLD(q_\kappa(\upsilon|X) || p_\theta(\upsilon|X)) \tag{7}$$
$$= \mathbb{E}_{q_\kappa(\upsilon|X)} \left[ \log \frac{q_\kappa(\upsilon|X)}{p_\theta(\upsilon|X)} \right]$$
$$= \mathbb{E}_{q_\kappa(\upsilon|X)} \left[ \log q_\kappa(\upsilon|X) - \log p_\theta(\upsilon, X) + \log p_\theta(X) \right]$$
$$= - L(\theta, \kappa; X) + \log p_\theta(X), \tag{8}$$

*Since $\log p_\theta(X)$ is independent of $q_\kappa(\upsilon|X)$, minimizing Equation equation 7 is equivalent to maximize $L(\theta, \kappa; X)$ in Equation equation 8.*

*Based on Bayes's theorem, we have:*

$$
\begin{aligned}
L(\theta, \kappa; X) &= \mathbb{E}_{q_\kappa(v|X)} \left[ -\log q_\kappa(v|X) + \log p_\theta(v, X) \right] \\
&= \mathbb{E}_{q_\kappa(v|X)} \left[ -\log q_\kappa(v|X) + \log \left( p_\theta(X|v) p_\theta(v) \right) \right] \\
&= \mathbb{E}_{q_\kappa(v|X)} \left[ \log \frac{p_\theta(v)}{q_\kappa(v|X)} + \log p_\theta(X|v) \right] \\
&= \mathbb{E}_{q_\kappa(v|X)} \left[ \log p_\theta(X|v) \right] - KLD \left( q_\kappa(v|X) || p_\theta(v) \right),
\end{aligned}
$$

*Under the assumption that $X$ is i.i.d. sampled from $\mathcal{D}_X$, we obtain the evidence lower bound (ELBO) objective:*

$$
\max \mathbb{E}_{X \sim \mathcal{D}_X} \left[ \mathbb{E}_{q_\kappa(v|X)} \left[ \log p_\theta(X|v) \right] - KLD \left( q_\kappa(v|X) || p(v) \right) \right].
$$

**Theorem 2** *The tractable ELBO of state-action distributional variational inference is:*

$$
\mathbb{E}_{X \sim \mathcal{D}_X} \left[ \mathbb{E}_{q_\kappa(v|X)} \left[ \sum_{i=1}^{N} \log p_\theta \left( s^{(i)}|v \right) + \log p_\theta \left( a^{(i)}|v, s^{(i)} \right) \right] + KLD \left( q_\kappa(v|X) || p(v) \right) \right] \quad (9)
$$

**Proof A.2** *Under the assumption of i.i.d. sampling on $X$, the probability of $q_\kappa(v|X)$ and $p_\theta(X|v)$ can be estimated via likelihood:*

$$
q_\kappa(v|X) = \prod_{i=1}^{N} q_\kappa \left( v|s^{(i)}, a^{(i)} \right) \quad (10)
$$

$$
p_\theta(X|v) = \prod_{i=1}^{N} p_\theta \left( s^{(i)}, a^{(i)}|v \right) = \prod_{i=1}^{N} p_\psi \left( s^{(i)}, a^{(i)} \right) p_\theta(\psi|v), \quad (11)
$$

*where $\psi$ denotes the parameters of distribution $p_\psi$.*

*In the reinforcement learning scenario, the action is sampled conditionally on the state, thus the posterior $p_\theta$ can be separated by:*

$$
\begin{aligned}
&p_\theta \left( s^{(i)}, a^{(i)}|v \right) \\
=& p_\theta \left( a^{(i)}|v, s^{(i)} \right) p_\theta \left( s^{(i)}|v \right) \\
=& p_{\psi_a} \left( a^{(i)} \right) p_\theta \left( \psi_a|v, s^{(i)} \right) p_{\psi_s} \left( s^{(i)} \right) p_\theta \left( \psi_s|v \right), \quad (12)
\end{aligned}
$$

*where $\psi_s$ and $\psi_a$ denote the decoded parameters of the distribution. Then the trac ELBO objective can be written as:*

$$
\begin{aligned}
&\mathbb{E}_{X \sim \mathcal{D}_X} \left[ \mathbb{E}_{q_\kappa(v|X)} \left[ \log p_\theta(X|v) \right] - KLD \left( q_\kappa(v|X) || p(v) \right) \right] \\
=& \mathbb{E}_{X \sim \mathcal{D}_X} \left[ \mathbb{E}_{q_\kappa(v|X)} \left[ \sum_{i}^{N} \log p_\theta \left( x^{(i)}|v \right) \right] - KLD \left( q_\kappa(v|X) || p(v) \right) \right] \\
=& \mathbb{E}_{X \sim \mathcal{D}_X} \left[ \mathbb{E}_{q_\kappa(v|X)} \left[ \sum_{i=1}^{N} \log p_\theta \left( s^{(i)}|v \right) + \log p_\theta \left( a^{(i)}|v, s^{(i)} \right) \right] + KLD \left( q_\kappa(v|X) || p(v) \right) \right].
\end{aligned}
$$

*Since $q_\kappa \left( v|s^{(i)}, a^{(i)} \right)$ can be modeled with Gaussian distribution, the results are also a Gaussian distribution Rakelly et al. (2019). Besides, $p_\theta \left( a^{(i)}|v, s^{(i)} \right) = p_{\psi_a} \left( a^{(i)} \right) p_\theta \left( \psi_a|v, s^{(i)} \right)$ and $p_\theta \left( s^{(i)}|v \right) = p_{\psi_s} \left( s^{(i)} \right) p_\theta \left( \psi_s|v \right)$, where $\psi_s$ and $\psi_a$ denote the decoded parameters of the distribution. Thus, for any differentiable $p_{\psi_s}$ and $p_{\psi_a}$, the ELBO objective is tractable.*

In summary, SA-DAE has a three-step pipeline to minimize the objective of Equation equation 9:

1. Sample a batch of $X$ from the dataset $\mathcal{D}_X$;

2. Infer latent code $v$ via the likelihood probability based on Equation equation 10;

3. Compute the reconstructed log-probability of state-action pair based on Equation equation 12 and Equation equation 11, and compute the KL divergence between posterior and prior of $v$, then apply the gradient with respect to $\kappa$ and $\theta$.

# B EXPERIMENTAL SETTINGS

## B.1 MULTIPLE-DOMAIN MULTIPLE-CUSTOMER LONG-TERM SATISFACTION PROBLEM (LTS)

Long-term satisfaction (Choc/Kale) problem comes from a synthetic environment in the Google RecSim framework Ie et al. (2019). In this environment, the recommendation system sends items of content to customers, and the goal is to maximize customers' engagement in multiple timesteps. The items of content are characterized by the score of clickbaitiness. The engagement of customers is determined by the clickbaitiness score of content and the long-term satisfaction score. The higher clickbaitiness score leads to a larger engagement directly but leads to a decrease in the long-term satisfaction while the lower clickbaitiness score increases satisfaction but leads to a smaller engagement directly. Moreover, the long-term satisfaction is a coefficient to rescale the engagement of the given item of content. In particular, the value of engagement for customer $i$ at timestep $t$ is sampled from a Gaussian distribution $\mathcal{N}\left(\mu_t^i, \sigma_t^{i^2}\right)$, which is parameterized by

$$\mu_t^i := \left(a_t^i \mu_c^i + \left(1 - a_t^i\right) \mu_k^i\right) SAT_t^i$$
$$\sigma_t^i := \left(a_t^i \sigma_c^i + \left(1 - a_t^i\right) \sigma_k^i\right),$$

where $i$ denotes the index of customer, $a_t^i$ denotes the clickbaitiness score of the given item. $\mu_c^i$, $\mu_k^i$, $\sigma_c^i$ and $\sigma_k^i$ are hidden states of the customer $i$. $SAT_t^i$ denotes the long-term satisfaction score, which is updated by $a_t$:

$$SAT_t^i := sigmoid\left(h_s^i \times NPE_t^i\right)$$
$$NPE_t^i := \gamma^i NPE_{t-1}^i - 2\left(a_t^i - 0.5\right),$$

where $NPE_t^i$ denotes the net positive exposure score of the customer $i$, $\gamma^i$ denotes the memory discount of $NPE_t^i$, and $h_s^i$ denotes the sensitivity ratio of $NPE$ to satisfaction. $\gamma^i$ and $h_s^i$ are also hidden states in this environment.

For the LTS environment, the hidden states $\mu_c^i$, $\mu_k^i$, $\sigma_c^i$, $\sigma_k^i$, $h_s^i$ and $\gamma^i$ define customer behaviors. To construct the problem as the multiple-domain multiple-customer environment, we use $\mu_c^i$ to represent the parameter of group information $z_d$, which are the same among customers in the same simulator, and the rest of the hidden states are randomly sampled for each customer, to represent the parameter of $z$.

In our scenario, group-behavior parameters can only be identified through multiple customers' trajectories via an unkown projection function $f$, i.e., $z_d^t = f(\mathcal{T}_{0:t})$, where $\mathcal{T}_{0:t} = [S_0, A_0, ...S_{t-1}, A_{t-1}, S_t]$, and $S$ and $A$ denote the state and action sets of all customers in the same domain. To model this scenario in the synthetic environment, we need to design extra observations for each customer to infer $z_d^t$. We consider the simplest case: $z_d$ is time-invariant and can be identified by $S_t$ at any timestep $t$. In implementation, each customer has an extra fixed observation sampled from Gaussian distribution $o^i \sim \mathcal{N}(\mu_c, \omega_c)$. Then $z_d$ is identifiable through multiple customers' observation, since $z_d = \mathrm{E}_i\left[o^i\right]$.

We construct the simulator in which $\mu_c = 14$ as the test simulator and select the training simulators by equidistant sampling parameters $\mu_c$ from environment-context space and removing those near the testing simulator. The observation of customers is sampled at $\omega_c = 4$. Then we construct three tasks with different transfer difficulties via controlling the removed range of parameters from training sets $\{\rho_{train}\}$:

- LTS1: $\{\rho_{train}\} = \{\rho_{\mu_c} : |\mu_c - 14| \geq 2 \wedge 6 \leq \mu_c < 22, \mu_c \in \mathbb{N}\}$
- LTS2: $\{\rho_{train}\} = \{\rho_{\mu_c} : |\mu_c - 14| \geq 3 \wedge 6 \leq \mu_c < 22, \mu_c \in \mathbb{N}\}$
- LTS3: $\{\rho_{train}\} = \{\rho_{\mu_c} : |\mu_c - 14| \geq 4 \wedge 6 \leq \mu_c < 22, \mu_c \in \mathbb{N}\}$

The environment-context space is set to $\mu_c \in [6, 22)$, taking 14 as the middle point, and we take the number of 2, 3 and 4 as the removed range for LTS1, LTS2 and LTS3, respectively.

In distortion test, we follow the setting of LTS3 and select $\mu_k$ as the distored parameter. In particular, for all simulators, $\mu_k^i \sim 4 + U(-\beta, \beta)$. Without considering distortion, $\mu_k^i$ is set to 4 for all customers in the simulators.

### B.2 Driver Program Sequential Recommendation Problem

To demonstrate the effectiveness of the proposed method, we deploy OapRS in a real-world recommendation application, for a large-scale ride-hailing platform. The goal of the platform is to balance the demand from passengers and the supply of drivers, i.e., helping drivers finish more orders, and satisfying the more trip demand from passengers. The driver program recommendation (DPR) is one of the typical recommendation scenarios in the platform. In this scenario, to satisfy more demand from passengers, we would like to maximize the engagement of drivers via recommending reasonable items from the program. The engagement is characterized by the number of orders completed by each driver. The programs contain tasks for the driver to accomplish. If the driver completes the recommendation program, his/her engagement will increase. However, the difficulty of the task has influences on the completion rate since drivers respond differently to tasks, which also affects the effectiveness of recommendations. The responses are also changing for a driver over time. For example, the base number of demand is not in the same order of magnitude among the different cities, since the total passenger volumes vary. Hence, the drivers who have similar personas may behave differently in different cities, leading to diverse recommended programs.

The driver program recommendation can be modeled as an MDP. For simplification, we assume the influence among drivers can be ignored. It is reasonable since drivers almost have no ideas about other drivers' tasks. In the DPR environment, we regard each day as a timestep. At timestep $t$, the recommendation system policy $\pi$ sends a program $a_t^i = \pi(s_t^i)$ to driver $i$ based on the observed feature $s_t^i$. $a_t^i$ denotes the program features to characterize the action. The reward $r_t^i$ is the number of finished orders in the day. Our goal is to maximize the expectation of long-term engagement of drivers in different cities.

We divide real-world data into seventeen cities with two different periods for simulator learning. The Simulator is learned by technique Shang et al. (2019). Then the environment context is implied in the learned simulator models. In the semi-online test, we evaluate the policy performance with reality-gap via running the policy in the unseen city. The experiments allow testing the algorithms on the reality-gap through city and time period transfer, which is essentially environment context change.

## C Experimental details in Long-term Satisfaction Problem

We first train a policy given oracle environment context in the synthetic simulators, to test the effect of the environment contexts. Figure 6 shows the action distribution of optimal policies in different contexts of $\mu_c$. It can be seen that the larger $\mu_c$ leads to a larger percentage of the higher clickbaitiness-score item selected. Thus, a transferable policy should be aware of the differences of $\mu_c$ and adapt its decisions for better performance.

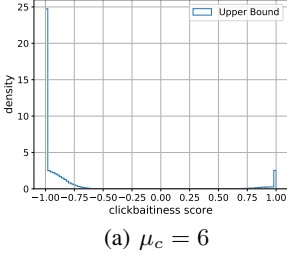
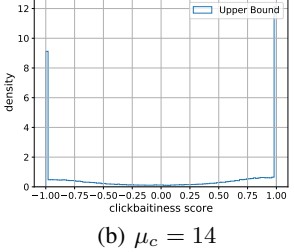
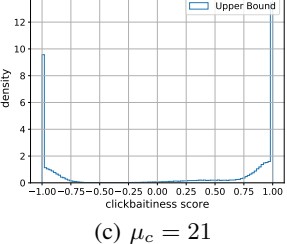

| (a) $\mu_c = 6$ | (b) $\mu_c = 14$ | (c) $\mu_c = 21$ |

Figure 6: Illustration of generative action distribution for optimal policies in LTS3. The score is rescaled into $(-1, 1)$ from $(0, 30)$. We sample 750 trajectories of different customers and draw the histogram of these sampled actions.

In the LTS environment, $z_d$ is only related to group state information $S$. Thus we train SA-DAE to reconstruct the state distribution instead of the state-action distribution. We draw 1000 customers for each simulator to the constructed state dataset $\mathcal{D}_X$. $q_\kappa \left( \upsilon | s^{(i)} \right)$ is a neural network which outputs the

Gaussian distribution parameters of $\upsilon$. The dimension of vector $\upsilon$ is 5 in this environment. We also model $p_\theta\left(\psi_s|\upsilon\right)$ with a neural network, which outputs the parameters of Gaussian distributions. The prior of $\upsilon$ is set to standard normal distribution, i.e., $p(\upsilon) = \mathcal{N}(0, 1)$.

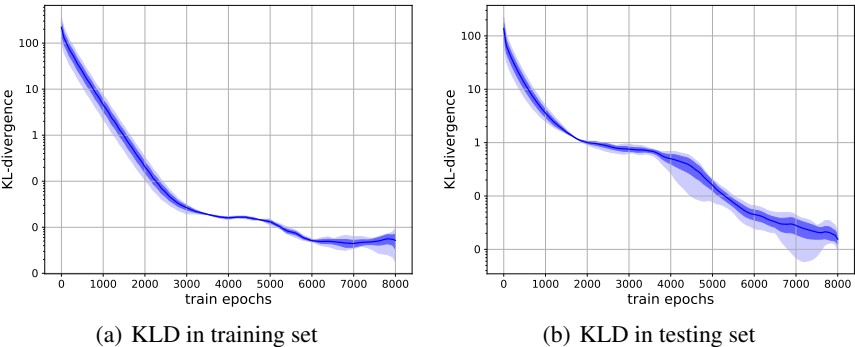

(a) KLD in training set

(b) KLD in testing set

Figure 7: Illustration of KL divergence of the training set and testing set in LTS3. The solid curves are the mean reward of three seeds. The dark shadow is the standard error, and the light one is the min-max range of three seeds.

We use KLD to measure the performance of reconstruction. Since $p_\theta\left(\psi_s|\upsilon\right)$ also outputs the parameters of Gaussian distribution, we compute the KLD directly via the analytic expression of Gaussian distribution between $p_\theta\left(s|\upsilon\right)$ and $\mathcal{N}(\mu_c, \omega_c)$. We test the KLD every 100 epochs. Figure 7 shows that the KLD in the testing set finally converges to the range of 0.01 to 0.02. Figure 8 shows the reconstruction distribution is also correlated.

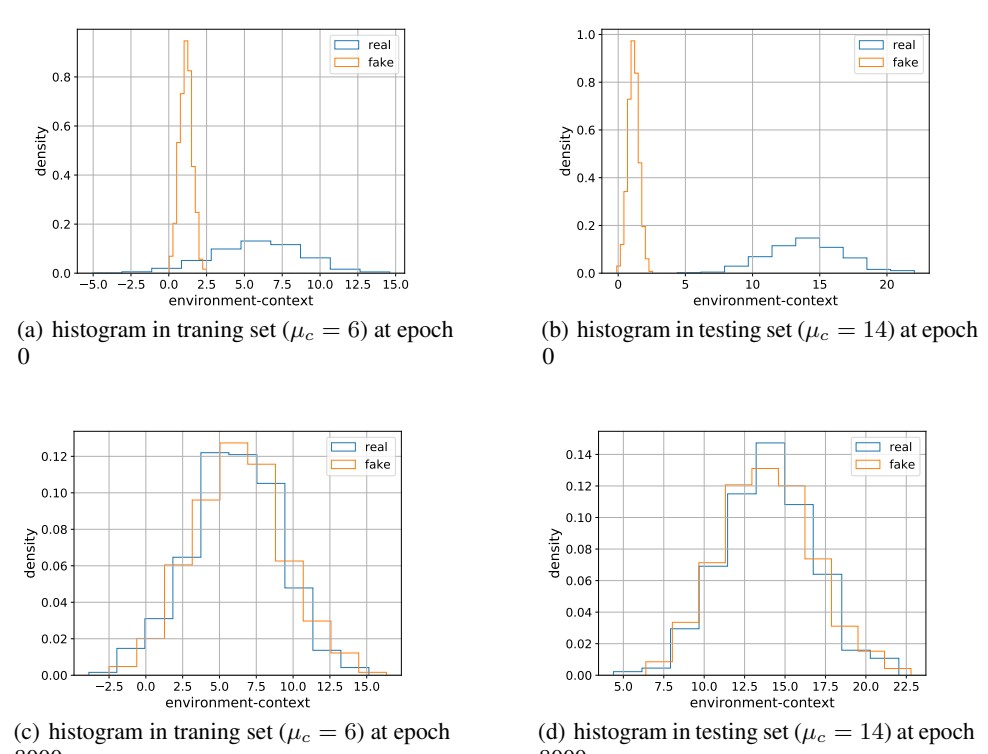

(a) histogram in traning set ($\mu_c = 6$) at epoch 0

(b) histogram in testing set ($\mu_c = 14$) at epoch 0

(c) histogram in traning set ($\mu_c = 6$) at epoch 8000

(d) histogram in testing set ($\mu_c = 14$) at epoch 8000

Figure 8: Illustration of the histogram about customer feature of $o^i$ in reconstructed and real data in the task of LTS3.

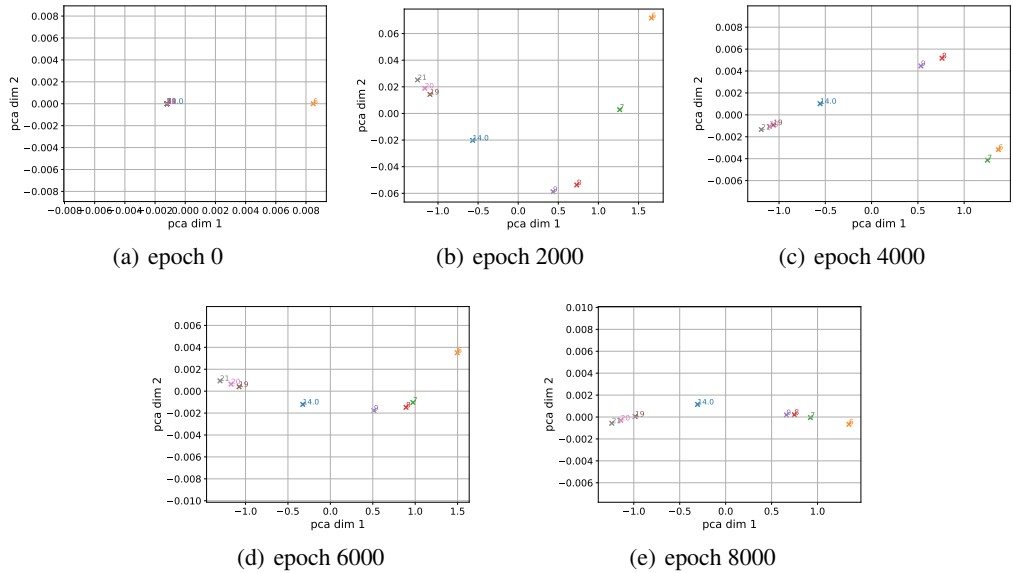

Figure 9: Illustration of the visualization on $\upsilon$. The X-axis denotes the first principal component, and the Y-axis denotes the second one. Each cross point denotes the projection of the latent code for the state distribution. The numbers with the same color to the point denote the ground-truth of $\mu_c$. Although $q_\kappa(\upsilon|X)$ is a Gaussian distribution, we only draw the mean of the distribution for legibility.

To evaluate the embedding performance of SA-DAE, we project $\upsilon$ into two-dimensional vectors via principal component analysis (PCA) Wold et al. (1987). We show the result of PCA in Figure 9. It can be seen that, after 6000 epochs, the latent code can be represented by the first principal component. And the value of $\mu_c$ linearly depends on the first principal component.

We finally test the robustness of environment-parameter extractor to the noise scale $\omega_c$. The result can be seen in Figure 10. The results show that the environment-parameter extractor can also work well in large noise $\omega_c$ situations.

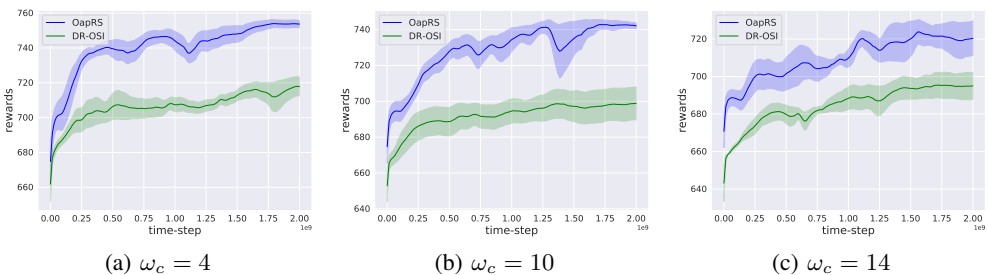

Figure 10: Illustration of the performance in synthetic environments in different $\omega_c$. The solid curves are the mean reward and the shadow is the standard error of three seeds.

# D    PPO IMPLEMENTATION OF OAPRS

We use PPO Schulman et al. (2017) as an example and any policy-based methods (i.e., TRPO Schulman et al. (2015), SAC Haarnoja et al. (2018)) can also be plugged into the policy learning part. The pseudocode of OapRS is summarized in Algorithm 2.

---

**Algorithm 2** OapRS implemented with PPO

---

**Input**:

Offline expert data $\mathcal{D}$;

LSTM network $\phi_\varphi$ as environment-context extractor, parameterized by $\varphi$

Adaptive policy network $\pi_\theta$ and value function network $V_\theta$, parameterized by $\theta$

Model learning algorithm $G$; Rollout horizon $H$; minibatch $m$ for policy training

Training epochs $M$

**Process**:

1: Divide the offline dataset into $N$ partitions $\{\bar{\mathcal{D}}\}$ and generate $M \times N$ dynamics model $\{\rho\}$ by ensemble of $M$ model learning technique DEMER (Shang et al., 2019) $G$.
2: Pretrain the state-action distributional embedding model $q_\kappa(\upsilon|X)$ on $\{\bar{\mathcal{D}}\}$.
3: **for** 1, 2, 3, ... **do**
4:     Random select the dynamics model $\rho^i$ in $\{\rho\}$ and sample $s_1$ from $\bar{\mathcal{D}}$ which is used to train $\rho^i$
5:     initialize buffer $\mathcal{D}_{rollout}$
6:     **for** $t=1,2,...,H$ **do**
7:         Sample $z_t$ from $\phi(z_t|\rho_t^i)$ and then sample $a_t$ from $\pi(a_t|s_t, z_t)$
8:         rollout one step $s_{t+1}, r_{t+1} \sim \rho_t^i(s_t, a_t)$ and then add $(s_{t+1}, r_{t+1}, s_t, a_t, z_t)$ to $\mathcal{D}_{rollout}$
9:         **if** $t \mod m == 0$ **then**
10:             Compute the long-term reward estimates, $\hat{R}_t^{\pi_{\theta_m}}$, for each time-step $t$.
11:             Compute the advantage estimate, $\hat{A}_t^{\pi_{\theta_m}}$ based on the value function $V_\theta(s, z)$, where $z$ is the output of extractor $\phi_\varphi$.
12:             Compute the gradient $\nabla_\theta$ and $\nabla_\varphi$ of the PPO objective :

$$\arg\max_{\theta,\varphi} \frac{1}{|\mathcal{D}_m|T} \sum_{\mathcal{T} \in \mathcal{D}_m} \sum_{t=0}^{T} q_\kappa(\upsilon_t|X_t) \sum_{s_t, a_{t-1}, a_t, z_t \in \mathcal{T}} L_{PPO}(\theta, \varphi, \upsilon_t, s_t, a_t, a_{t-1}, z_t).$$

13:             Apply $\nabla_\theta$ and $\nabla_\varphi$.
14:             Update value function by one-step gradient of the regression loss:

$$\arg\min_{\theta} \frac{1}{|\mathcal{D}_m|T} \sum_{\mathcal{T} \in \mathcal{D}_m} \sum_{t=0}^{T} q_\kappa(\upsilon_t|X_t) \sum_{s_t, a_{t-1} \in \mathcal{T}} \left( V_\theta(s_t, z_t')\phi_\varphi(z_t'|z_{t-1}', s_t, a_{t-1}, \upsilon_t) - \hat{R}_t^{\pi_{\theta_m}} \right)^2$$

15:             Update the state-action distributional embedding model $q_\kappa(\upsilon|X)$ on $\mathcal{D}_{rollout}$
16:             $\mathcal{D}_{rollout} \leftarrow \emptyset$
17:         **end if**
18:     **end for**
19: **end for**

---

## E EXPERIMENTAL DETAILS IN REAL-WORLD RIDE-HAILING PLATFORM

We collect offline data from the platform and construct our real-world simulators by imitation from data Shang et al. (2019). The results show that the order completion in the reconstructed simulators is generally consistent with the real-world data.

We train the SA-DAE in the training set and test the reconstructed data distribution in the unseen environment. The training dataset $\mathcal{D}_X$ comes from human expert data in the training set. $q_\kappa \left( \upsilon|s^{(i)}, a^{(i)} \right)$ is a neural network which outputs the Gaussian distribution parameters of $\upsilon$. The dimension of the vector $\upsilon$ is 200 in this environment. $p_\theta \left( \psi_a|\upsilon, s^{(i)} \right)$ and $p_\theta \left( \psi_s|\upsilon \right)$ are modeled with neural networks, which output the parameters of the distributions. The action reconstruction is modeled with Gaussian distribution since it is continuous in the environment. However, the state space includes continuous and discrete features. For simplification, we assume the continuous features are independent of discrete features. Thus we model them with Multivariate Gaussian distribution and categorical distribution respectively. The prior of $\upsilon$ is set to standard normal distribution, i.e., $p(\upsilon) = \mathcal{N}(0, 1)$.

We use KLD to measure the reconstruction performance. Since the dimension of state-action space is high and the distribution is complex, we use Kernel Density Estimation (KDE) Rosenblatt (1956) to estimate the probability density function (PDF) of real and reconstructed data. Then the KLD of two

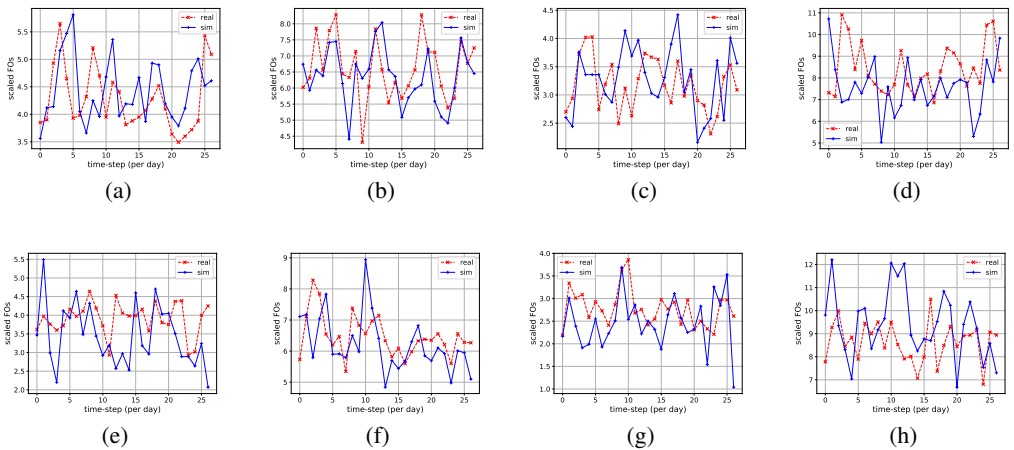

Figure 11: Illustration of the daily finished-order curves of the constructed simulator and real data. Each column shows the results of the same city. Each row shows the results in the same period.

datasets is computed based on it. In particular,

$$KLD(\mathcal{D}_a, \mathcal{D}_b) = \frac{1}{||\mathcal{D}_a||} \sum_{x \in \mathcal{D}_a} \log \frac{f_a(x)}{f_b(x)}, \tag{13}$$

where $||\mathcal{D}_a||$ denotes the number of samples in the dataset, and $f_a$ and $f_b$ denote the PDF of real and reconstructed data estimated by KDE. We test the KLD every 100 epochs. Figure 12(a) shows that the KLD between the real data $X$ and the reconstructed distribution $p_\theta(X|v)$ steadily converges to 0.6, which demonstrates nontrivial reconstruction performance. Figure 15 shows histograms for examples of real and reconstructed data on a single feature, which are also significantly correlated.

To evaluate the embedding performance of SA-DAE, we performed the hidden state prediction experiments Akkaya et al. (2019). If the embedding variables store useful information about the distribution, for a simple neural network, the KLD prediction error between arbitrary two datasets would be negatively correlated with the training epochs. We use another one-layer neural network to predict the KLD of two data pairs $(X_i, X_j)$, given their embedding variable $(v_i, v_j)$. The neural network has one 32-unit hidden layer with tanh as the activation function and links to a linear layer to predict the KLD computed by Equation 13. The neural network is initialized and retrained for the same epochs, every 100 iterations of SA-DAE learning. Figure 12(b) shows the mean absolute error (MAE). The MAE has 26% improvement than the initial variable, which implies the embedding variable is helpful to infer the relation of two distributions.

## F  ADDITIONAL DETAILS FOR OAPRS

### F.1  MAPPING FUNCTION CONSTRUCTION

Given a specific SRS application, we can split the state space into the following parts:

- Customer-related static states: the states which are invariant to the recommended actions and are fixed for each customer. For example, the age and gender of each driver. Since we model the process of interaction between the platform and a customer to an MDP, the customer is fixed in a trajectory and thus customer-related states are invariant;

- Timestep-related static states: the states which are invariant to the recommended actions and are fixed for each time-step. For example, the weather for each day. We train and the dynamics model with the same time period as the collected dataset and thus the timestep-related states can be found from the dataset and fixed;

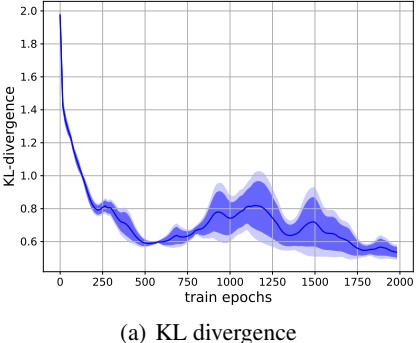

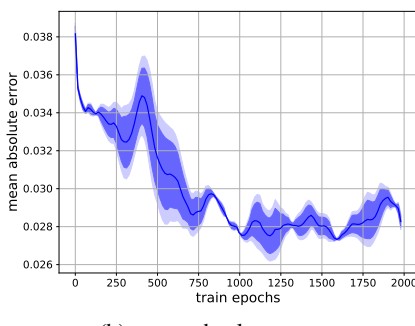

(a) KL divergence

(b) mean absolute error

Figure 12: Illustration of the reconstructed and embedding performance on SA-DAE. The solid curves are the mean reward of two seeds. The dark shadow is the standard error, and the light one is the min-max range of two seeds.

- Dynamics-related states: the states which are dependent on the given recommendation actions and hard to hand-code by rules. For example, the completed orders and GMV of a driver in a day given the predicted completed orders and the recommendation action;
- Rule-related states: the states which can be computed by rules given other states. For example, we can compute the cost of a driver in a day, given the completed orders and the recommendation action, and we can compute the averaged GMV of a driver by averaging the predicted completed orders of recent days. Instead of learning to predict all the states, the neural network is used to predict the "dynamics states".

After sampling an initial state $s_0$ from the dataset at the beginning, for each timestep $t$, the trajectories of the environment model can be sampled by the following steps:

1. Fix the customer-related static states: $s_t^{customer} = s_{t-1}^{customer}$;

2. Set the timestep-related static states $s_t^{timestep}$ with the state in dataset ;

3. Predict the dynamics states by neural network model: $s_t^{dynamics} = NN(s_{t-1}, a_{t-1})$, where $NN$ denotes the trained neural network;

4. Update the constructable states by rule:

$$s_t^{rule} = mapping(s_{t-1}, a_{t-1}, s_t^{timestep}, s_t^{dynamics})$$

5. Put them together: $s_t = [s_t^{customer}, s_t^{timestep}, s_t^{dynamics}, s_t^{rule}]$.

The framework of semi-dynamics model learning is general to other problems since it is easy for human experts to construct some rules for some states with a well-defined relationship and any number of rules can be embedded to reduce the complexity of learning. The more rules we have, the less learned states are needed. We list the different type of state of our application in Table 1.

Table 1: Different type of state in the state space.

| Category | Dimensionality | Description |
|---|---|---|
| $s^{customer}$ | 54 | Personas features |
| $s^{timestep}$ | 6 | The weather and holiday features et al. |
| $s^{dynamics}$ | 3 | The probability to drive in the day, the number of completed orders and the averaged GMV per order |
| $s^{rule}$ | 140 | The historical recommendation actions and the $s^{dynamics}$ in 7 days, some statistics and other construable features. |

### F.2 Augmentation techniques in OapRS

In SRS, there are several common choices to split the dataset: First is split by periods since the customer often behaves differently across periods in the applications. The second is split by some identification category features in the application since it often represents the domain classes. In our application, we split the dataset by the month and city id.

Some general ensemble techniques like learning with different initial neural network parameters, different hyper-parameters, different learning algorithms can also be used for dynamics model augmentation. However, we do not implement these methods for resource limitation and find the approach works in our application.

### F.3 Dynamics model performance evaluation

We split the dataset into a training set (80%) and a testing set (20%), the environment model is learned in the training set and evaluated in the testing set. Multiple metrics that are concerned by the operational staff can be used for evaluation. For example, the similarity of total completed orders of drivers in a city between the data run by the environment model and the corresponding real-world data. We use Pareto improvement (a new model where some metrics will gain, and no metrics will lose) of the metrics in the testing set to define the performance improvement of the environment model. The metrics we selected are the mean absolute percentage error (MAPE) of the following statistics: (1) the total cost for each day; (2) the total number of driver is driving for each day (i.e., the number of completed order is larger than zero); (3) the total GMV for each day.

### F.4 Detailed structure and optimization tricks

The network structure of OapRS is shown in Figure 13. In the implementation, we use two independent network structures for the value function and policy learning. Moreover, there is a skip-connection for the environment-aware layer to reuse the original input features. The embedding layer and environment-aware layer are modeled with Multilayer Perceptron (MLP). The environment-context extractor layer is modeled with a single-layer LSTM network Hochreiter & Schmidhuber (1997). We use the same network structure and hyperparameters in the two experiments, but the complexity of neural networks are different. Table 2 reports the detailed settings.

In the DPR environment, we model the distribution of continuous states with Multivariate Gaussian distribution. The covariance of the Multivariate gaussian distribution should be a symmetric positive semi-definite matrix. We construct the covariance matrix based on Cholesky decomposition Golub & Van Loan (1996). In particular, a symmetric positive-definite matrix $\Sigma$ is a decomposition of the form:

$$\Sigma = LL^\top,$$

where $L$ is a lower triangular matrix with real and positive diagonal entries, and $L^\top$ denotes the transpose of $L$. We use a neural network with the softplus activation function to output the elements of $L$, where the softplus is used to output a positive value. Then, we regard the matrix $LL^\top$ as the covariance of the Multivariate Gaussian distribution. The hyperparameters of SA-DAE in the two experiments are shown in Table 3.

In addition, we use several tricks in the training process of the environment-context extractor and adaptive policy:

**Larger entropy coefficient**   We set the entropy coefficient to 0.02 for the LSTM-based policy and 1e-5 for MLP-based policy in the two problems. We found that the smaller entropy coefficient leads to performance collapse after some iterations when training with LSTM-based policy.

**Multiple environments sampling**   We sample from all training environments uniformly for $\nabla J(\theta)$ computation every iteration. The policy updates would be unstable and inefficient if just sampling from a single environment.

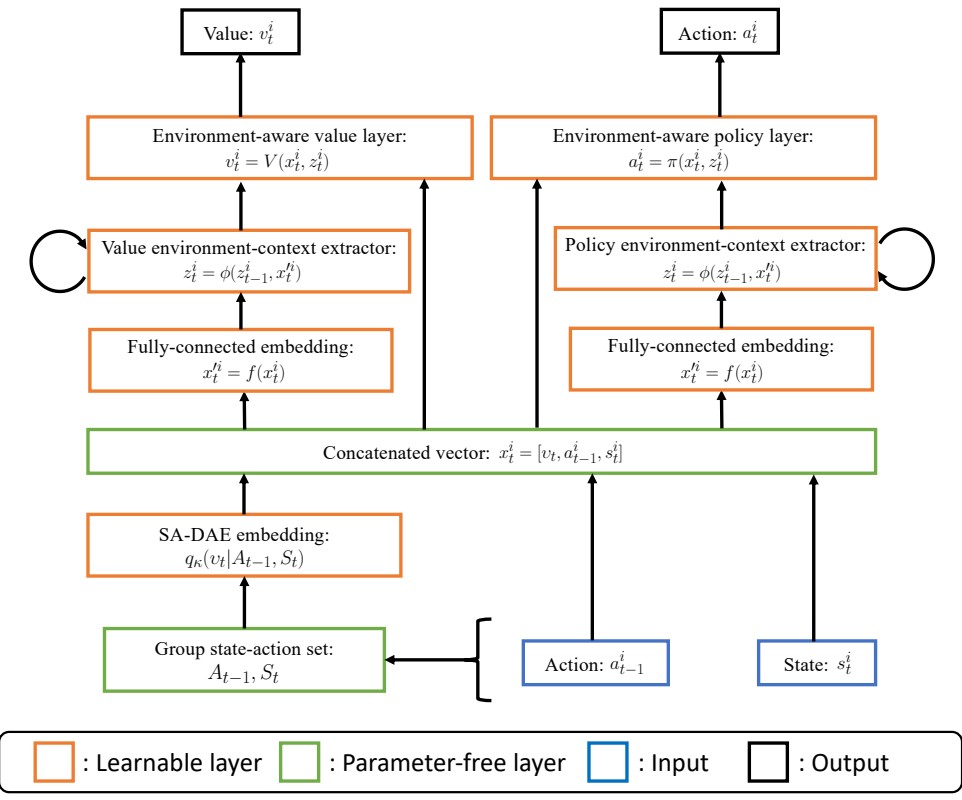

Figure 13: Illustration of the network structure for OapRS.

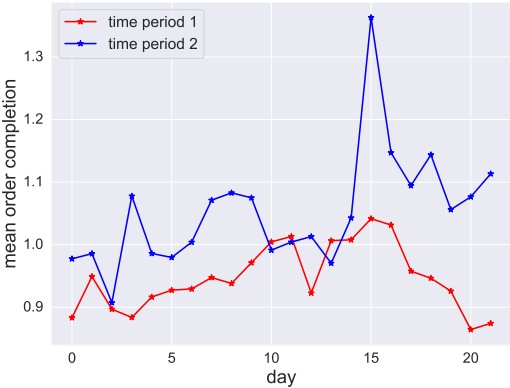

Figure 14: Illustration of the mean order completion in the real-world ride-hailing application. The number of mean order completion is computed by the mean value of the order completion in a city per day. The result is scaled with a constant value. The legend "time period 1" and "time period 2" denote the data collected in adjacent 22 days of the same city.

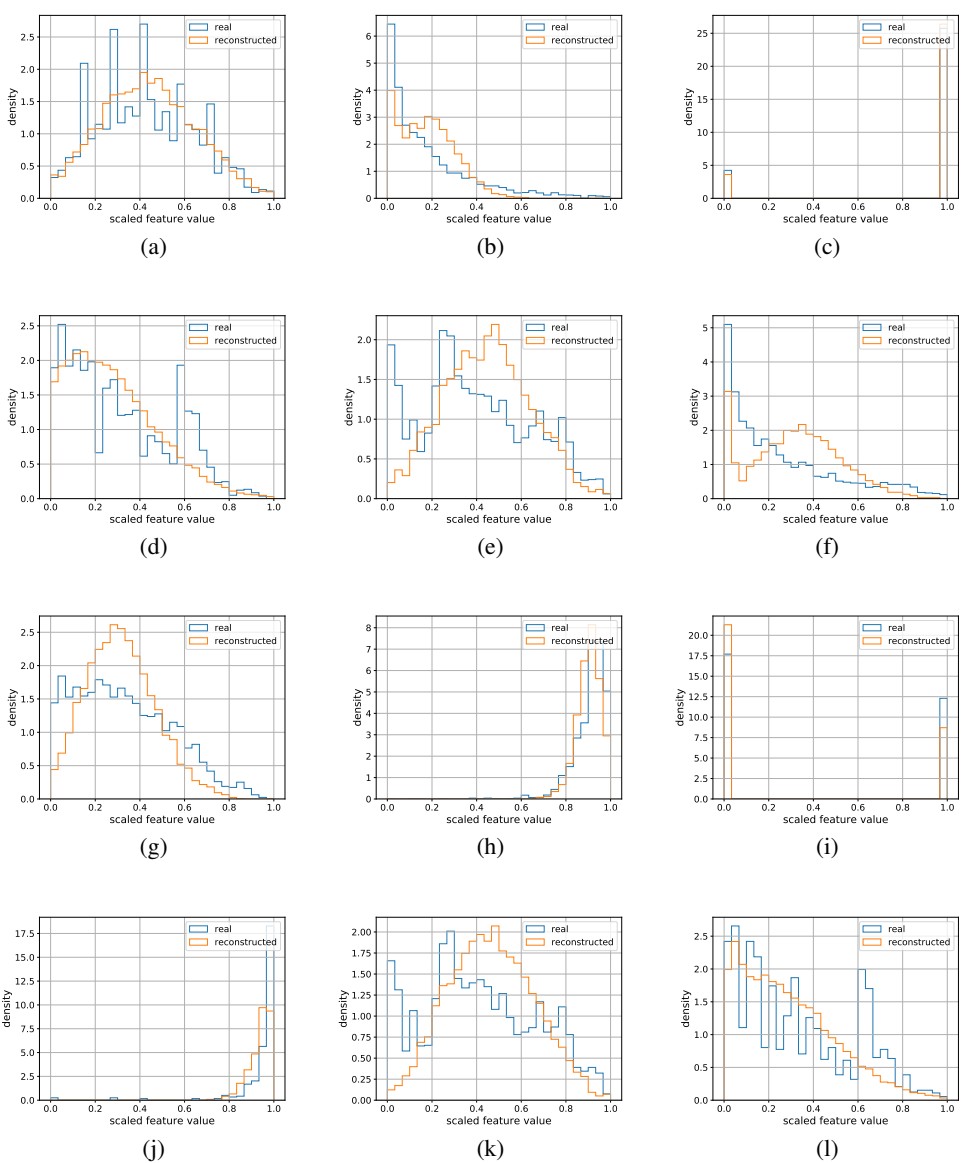

Figure 15: Illustration of the histogram about customer features of reconstructed and real data.

Table 2: The hyper-parameters of OapRS for policy learning.

| Hyperparameter | LTS | DPR |
|---|---|---|
| Discount factor $\gamma$ | 0.99 | 0.9 |
| Horizon $H$ | 140 | 30 |
| Entropy regularization coefficient | 0.02 | 0.001 |
| PPO clipping parameter $\epsilon$ | 0.2 | |
| Learning rate | from 1e-4 to 1e-6 | |
| Batch size | 30000 | 120000 |
| Sample reuse | 3 | |
| Optimizer | Adam | |
| L2 regularization weight | 1e-6 | |
| Activation function of hidden layer | leaky relu | |
| Activation function of policy output | tanh | |
| Fully-connected embedding $f$ | [128, 128, 128,32] | [512, 512, 256] |
| Unit of environment-context extractor $\phi$ | 64 | 256 |
| Environment-aware layer of $V$ and $\pi$ | [128, 64] | [512, 256] |

Table 3: The hyper-parameters of OapRS for SA-DAE.

| Hyperparameter | LTS | DPR |
|---|---|---|
| Learning rate: | 2e-5 | 1e-6 |
| Layer Normalization: | False | |
| Optimizer: | Adam | |
| L2 regularization weight | 0.1 | 0.001 |
| activation function of hidden layer | leaky relu | |
| Embedding layer $q_\kappa(v|s,a)$ | [512, 512] | |
| Reconstructed layer $p_\theta(\psi|v)$ | [512, 512] | |
| units of latent code | 5 | 200 |

Table 4: The averaged scaled daily rewards of real-world deployment in RDP environments

| | OapRS | MOPO |
|---|---|---|
| before | 0.9019 | 0.8857 |
| after | 0.9641 | 0.8865 |
| improvement (%) | 6.9% | 0.1% |

