# OpenReview forum: "Offline Adaptive Policy Leaning in Real-World Sequential Recommendation Systems"
_ICLR.cc/2021/Conference — Reject_

### Official Review · AnonReviewer4 · 2020-10-27
**The  paper extends domain randomization techniques from robotics to SRS, which can take care of non-stationarity (for free) on top of the usual modeling errors.**

**Rating:** 4
**Confidence:** 4

**Review:**

Paper proposes curtaining the impact of simulator not being faithful to reality in a few ways: (a) define a policy that takes environmental context as input, (b) define an environment context predictor, (c) training scheme that learns the adaptable policy as well as the context predictor. The idea is that the learned policy is adaptable to diverse simulators defined using offline data. So when it is deployed online, it can adapt to the dynamics of real world changing across time. They use sequential recommendation systems (SRS) as the application to demonstrate this.

The key idea of the paper seems to be to extend domain randomization techniques from robotics to SRS, and justify that it can additionally take care of non-stationarity (for free), on top of the usual modeling errors.


1. The inability to sample in the real-world maybe restrictive for robotics, but is a moot point for SRS. There is almost no cost for exploration (you will have to show recommendations anyway and the cost of showing bad recommendations is almost nil at medium-large scale) and augmenting your existing data should be much much easier. Is there compelling evidence otherwise?

>> solve an offline problem that policies can applied to real-world applications without any additional online sample.


2. Is there a quantitative reference for non-stationarity of consumer behavior and its impact on revenue loss or something similar? Kruger et al. has no real world examples itself. Even if it is non-stationary, in real world systems, policies are retrained at a high enough frequency that this should not be an issue.

3. The description before equation (1) seems wrong. If there is a \forall \rho, then equation (1) should have such a constraint. Taking expectation with respect to \rho is not considered robust. This misunderstanding decreases confidence in the validity of the results presented.


4. The paper is generally dense and hard to understand at several places. Some of the statements are imprecise (some words seem to be misused) and detract from the key contributions of this work.


 - For instance, the following should be rephrased and jargon should be minimized:

>> Hidden confounder factors (Shang et al., 2019) obstruct the deterministic predictions. As a result, the high-confidence regions are drastically reduced and thus the exploration of policy learning are obscured.

 - Unclear what this is aiming to say (without explicitly discussing how policy 'exploration' is constrained):

>> instead of constraining policy exploration in high-confidence regions

 - Unclear (what is inconsistent?):

>> in regions with inconsistency

 - Unclear. What is the two level structure below. Is the example with domains/countries supposed to be a proxy for environmental context?

>> the environments include a two-level structure: ...

 - Unclear (what is the special structure? how is the environment made context agnostic?):

>> and a special environment structure makes the environment context agnostic

 - Unclear how stochastic environments have an influence in policy optimization without fixing a learning algorithm:

>> However, in stochastic and non-stationary environments, the learned dynamics have high uncertainty and thus constrain the policy to exploration in a small region.

 - Unclear what the performance metric of each simulator is:

>> maximize the performance metric of each simulator

 - Repeated use of suboptimality and optimality before formally defining it: Also unclear what does "all possible transitions out of consistency region" mean:

>> to reach suboptimal policies (in Figure 1(c)), we learn to make optimal decisions for all possible transitions in
the states out of the consistency region


 - What is "exploitation" here, isn't this confounding/overloading an existing technical term:

>> learned policies tend to exploit regions where the model is distorted


 - What is a penalty and how is it consistent, and how does it obstruct? If it is relevant to differentiate the proposed solution with respect to this weakness of prior works, more information and clarity is needed.

>> Since the environment has stochasticity, learning by consistency penalties will obstruct the policy into a small region for exploration.

>> without relying on the consistency penalty

 - Unclear:

>> instead of constraining policy exploration in high-confidence regions

---

> ### Author Response · Authors · 2020-11-20
> **Reply to AnonReviewer4 (part 1)**
>
> We thank the reviewer for the constructive suggestions. We address the detailed concerns in the following.
>
> A: In SRS, there is almost no cost for exploration, and augmenting your existing data should be much much easier.
>
> In some scenarios, the wrong user recommendations indeed do not have a serious impact. However, in many budget-related recommendation system scenarios, unreasonable recommendations will generate unacceptable additional costs. Besides, on a large platform, any percentage points of extra meaningless spending correspond to huge economic losses. For example, in coupon sending scenarios, to encourage certain customer behavior (e.g., making purchases), we would like to give customers some bonuses or discounts on the price. We would like to learn the optimal bonus/discount decisions for each customer. In online RL, the process of exploration leads to a large number of unreasonable bonuses/discounts. A product purchased with a too large bonus/discount means meaningless cost, and the total cost is unpredictable and uncontrollable in face of a large number of trial-and-error. For the products with a low-profit margin, the cost of one product sold with an overly large bonus/discount requires significant additional sales to cover.
>
>
> B: Is there a quantitative reference for non-stationarity of consumer behavior and its impact on revenue loss or something similar?
>
> We update the reference for discussing non-stationarity in real-world SRS applications [1,2,3,4]. To show the non-stationarity of our environment, we have updated Figure 14 to the appendix to show the difference in the order completion behavior of the same drivers in two adjacent periods of a city. If the interval is longer, the difference could be more obvious.
>
>
> C: Even if it is non-stationary, in real-world systems, policies are retrained at a high enough frequency that this should not be an issue.
>
>
> High-frequency retraining may not always be feasible w.r.t. the rate of changes in the environment, due to training time and computational resources. It may also fall into the 'wild goose chasing' situation, where the policy is constantly trying to adapt but always lagging behind the changes in the environment. We believe that learning an adaptive policy is a more principled approach and eventually pays off. Besides, the retrain pipeline is also compatible with the adaptive policy approach, since we still can update our model with additional online feedback data. The ability to adapt increases the tolerability of policy to the delay of retraining.
>
>
> [1]  Xiangyu Zhao, Changsheng Gu, Haoshenglun Zhang, Xiaobing Liu, Xiwang Yang, and Jiliang Tang.   Deep reinforcement learning for online advertising in recommender systems.
>
> [2] Philip S. Thomas, Georgios Theocharous, Mohammad Ghavamzadeh, Ishan Durugkar, and Emma Brunskill.   Predictive off-policy policy evaluation for nonstationary decision problems,  with applications to digital marketing.
>
> [3] Shi-Yong Chen, Yang Yu, Qing Da, Jun Tan, Hai-Kuan Huang, and Hai-Hong Tang.  Stabilizingreinforcement learning in a dynamic environment with application to online recommendation.
>
> [4] Chang Li and Maarten de Rijke. Cascading non-stationary bandits: Online learning to rank in thenon-stationary cascade model.

---

> ### Author Response · Authors · 2020-11-20
> **Reply to AnonReviewer4 (part 2)**
>
>
> D: The description before equation (1) seems wrong. If there is a $\forall \rho$, then equation (1) should have such a constraint. Taking expectation with respect to $\rho$ is not considered robust.
>
> We thank the review for the rigorous review of the equation. Although "$\forall \rho$" does not appear in equ (1), $J_\rho$ has included $\phi(z_t|\rho_t)$ to identify $\rho$. Therefore, theoretically, it equivalent to optimize “$J_\rho (\pi_\phi), \forall \rho$" and "$E_{\rho \sim \mathcal{T}} [J_\rho (\pi_\phi)]$" if $P[\rho] > 0, \forall \rho
> \in \mathcal{T}$, which is correct by uniform sampling in our algorithm. We use the expectation of dynamics set to match the process of our algorithm. We add the condition "$P[\rho] > 0, \forall \rho \in \mathcal{T}$" and keep the expectation in the paper.
>
> Considering the $\phi(z_t|\rho_t)$ in the expectation, equ (1) is to maximize the performance of policy for each $\rho \in \mathcal{T}$. Ideally, if the deployment environment $\rho_r$ is in the dynamics set $\mathcal{T}$, the policy can make optimal decisions in the environment. Empirically, if the distribution of dynamics set $\mathcal{T}$ can cover the deployment environment $\rho$, the policy can make robust decisions relying on the generalization ability of neural network. Then we discuss an efficiency dynamics set construction method for better covering the deployment environment (imitate dynamics from the dataset with ensemble techniques).
>
>
> E: Unclear on the meaning of high-confidence/inconsistent region, exploration constrained, and suboptimality of policy on traditional offline model-based method.
>
> The discussion of high-confidence/inconsistent/uncertainty region, exploration constrained and suboptimality of policy can be found in the mentioned previous offline model-based algorithms: MOPO [1] and MOReL [2]. In the following we explain the jargon based on MOPO:
>
> Tianhe et al  [1] propose a lower bound of the true performance of a policy $\pi$ with respect to an estimated dynamics model $\hat \rho$ and an admissible error estimator $u$ as uncertainty-penalized rewards (see equ (7)).  The uncertainty is to define the difference between the estimated dynamics model and the oracle dynamics (see equ (6)).
> By reward penalty with uncertainty, the policy is optimized in the region where the uncertainty is small than $\delta$. In our paper, we call the region "high-confidence/consistency region". The reward penalty avoids exploiting regions with low confidence of model prediction for better lower bound performance in real environment $\rho_r$, but implicitly constrain the exploration and lead to suboptimality of policy in the learned dynamics model $\hat \rho$ with respect to the original reward function. The formal description can be seen in section 4.1 and section 4.2 of MOPO.
>
> The uncertainty estimation is often done by the ensemble technique. In the deterministic environment, uncertainty can be used to estimate the confidence of the prediction. However, in the environment with stochasticity, the uncertainty not only comes from the predicted error, but also the stochasticity of the next states itself.
>
> F: What is the two-level structure? Is the example with domains/countries supposed to be a proxy for environmental context? What is the special structure? how is the special structure environment made context agnostic?
>
> The recommendation policy interacts with customers. Each customer can be regarded as an environment. Different customers are environments with different environment-parameters. However, the domain in which the customer belongs to affects the behavior of the customer. To represent the user behavior, we should not only consider the customer's trajectory but also the domain information for each timestep.
>
> We again thank the reviewer for remarking on the preciseness. We have updated the paper based on the suggestions.
>
>
>
> [1] Tianhe Yu, Garrett Thomas, Lantao Yu, Stefano Ermon, James Zou, Sergey Levine, Chelsea Finn,and Tengyu Ma. MOPO: model-based offline policy optimization.CoRR, abs/2005.13239, 2020.
>
> [2] Rahul Kidambi, Aravind Rajeswaran, Praneeth Netrapalli, and Thorsten Joachims. MOReL : Model-based offline reinforcement learning.CoRR, abs/2005.05951, 2020.

---

### Official Review · AnonReviewer1 · 2020-10-29
**A divide-and-conquer method for off-policy reinforcement learning based on model identification**

**Rating:** 4
**Confidence:** 4

**Review:**

This paper studies off-policy reinforcement learning for sequential recommendation. The basic idea is to summarize each possible environment dynamic (i.e., the state transition) into an environment context vector, and optimize policy with respect to this context vector accordingly. The proposed solution is evaluated on a simulated sequential recommendation environment, and a real-world ride-hailing platform, which considerably increases the creditability of the proposed solution.

The problem setup is a mix, which affects the generalization of the proposed solution. On the one hand, the solution is general for off-policy reinforcement learning, as I do not find any strong dependency of the proposed problem setup (i.e., a MDP) and the target sequential recommendation problem. However, on the other hand, the details in the proposed solution are so specific, which makes me question the generalizability of the proposed solution. For example, the reward function is assumed to be given by human experts. But in the experiments, it seems it is just the reward is defined by human expert, e.g., in the ride-hailing platform the reward is defined as the number of completed orders, rather than the reward function, e.g., how the state and action pair maps to a numerical reward. A clarification is needed here. And do we assume any non-stationarity in the reward function at all? There are other manually defined heuristics to “reduce model learning complexity”, e.g., “a hand-coded function $map(s’|\bar{s},s,a)$ to construct the left of the states”. And actually I did not find the details about this hand-coded function in Appendix F. If even the details were provided, I am not sure how to construct this mapping for new problems, e.g., beyond the ride-hailing problem, in general.

The key idea behind the proposed solution is to exhaust possible environment models from offline data, and estimate the policy for each estimated environment model. The environment models are created by splitting the offline data into non-overlapping partitions, and use a particular environment model learning algorithm (e.g., with different hyper-parameters) to generate different environment models. Hence, how to partition the offline data and how to estimate the environment model become vital; but there were not enough discussion or experiments on these two important aspects. The mentioned solution for dataset partitioning is “by domain knowledge” and environment model learning is by existing algorithms. And I also did not find any experiment studying how these two factors affect the proposed solution.

An autoencoder-based method is used for environment model identification, but I cannot exactly follow how the autoencoder informs the construction of environment-context extractor $\phi$? For example, how should I understand this sentence: “the extractor $\phi$ can infer environment-context both with $v$ and $\tau$”?

By the way, in Algorithm 2, what is the purpose behind line 19-22? Why do we need to generate those new episodes from the learnt environment model?

**Response after author rebuttal period**
I highly appreciate the detailed explanations and discussions the authors have provided during this period. It indeed clarifies my concerns and helps me better understand the setting and proposed solution. However, my major concern about the generalizability of the proposed solution still remines, as there are too many design choices depending on domain knowledge. I would keep my original recommendation; and if the paper could be accepted, I would like to encourage the authors to discuss the limitations of the proposed solution.

---

> ### Author Response · Authors · 2020-11-20
> **Reply to AnonReviewer1**
>
> We thank you for the careful review and the acknowledgment of our work. We address the detailed concerns in the following.
>
> We first clarify that the contribution we would like to claim is focusing on the offline policy learning problem in the sequential recommendation system (SRS) scenario. Compared with MuJoCo environments in which the dynamics models are static and deterministic, the sequential recommendation systems in the real-world applications introduce non-stationary and stochasticity. We consider the properties when learns the policy offline and propose to learn to adapt offline via an environment-parameter extractor. To learn the extractor in SRS, we formulate the environment context representation problem and propose to use a group-information embedding network (SA-DAE) with an RNN to learn the extractor.
>
> In the view of an SRS offline policy learning solution, the reviewer points out three important problems in the process of the algorithm: (1) how to construct a mapping function in general?（2）how to partition the offline data to exhaust possible environment models? （3）how to estimate the performance of the learned environment models? In fact, it is hard to give a general method to solve these problems. However, given a specific application, we can solve the problems with some general principles. We have answered these problems in the section "The details from training to the deployment process" of the "reply to common questions". We are adding the details of these implementations to the appendix F so as to guide other researchers to design their recommendation system. We will update the paper as soon as it is available.
>
> We also agree that the paradigm of learning to adapt may be a general solution in the general offline RL scenario, but the effectiveness has not been evaluated yet, and it is valuable to try in our future work.
>
> Response to the other detailed comments:
>
> A. In the ride-hailing platform the reward is defined as the number of completed orders, rather than the reward function. And do we assume any non-stationarity in the reward function at all?
>
> As mention in section 1 of the "reply to common questions", the number of completed orders is one dimension of our environment model. Given $s_t$ and $a_t$, the neural network predicts the completed orders of next state $CO_{t+1}$ for each driver firstly. Then the reward is $\alpha \times CO_{t+1}$, where $\alpha$ here denotes the rescaling coefficient. And thus the reward function does have non-stationarity since the ensemble dynamics models (especially those learned with datasets collected from different time periods) may predict different completed orders given the same state and action.
>
> B: The relation between the group-information auto-encoder and the environment-context extractor.
>
> The auto-encoder is used to extract the features of multiple customers. To learn the environment parameters, we concatenate the embedding vector of the auto-encoder and a single customer's state as the input vector to the environment-parameter extractor. The detailed structure can be seen in Figure 13.
>
> C: what is the purpose behind lines 19-22?
>
> These lines should not be included and has been deleted in the revision version paper.

---

### Official Review · AnonReviewer3 · 2020-10-30
**Novel idea to tackle with stochasticity and non-stationarity in RL for Recommender systems when learning using offline data.**

**Rating:** 7
**Confidence:** 2

**Review:**

The paper presents an adaptive model-based off-policy learning algorithm with applications to Sequential Recommender Systems (SRS) in mind. The novelty in the work is that it extends on other model-based and adaptive algorithms to suit the needs of SRS where stochasticity and non-stationarity are a part of the problem--this any learned model has to be robust to distortion because of off-policy setting. The key idea is to construct an 'environment-parameter extractor' to identify the dynamics system to adapt at runtime.

The background and notations in Section 3 and 4 are well written and rigorous, however, a little more grounding to a sequential recommender system setting would help in two ways: make the paper more self-sufficient for a Recommender systems audience, and also motivate the distinction between other model-based and other adaptive policy learning methods.

In the experiments section, it is worth discussing what is the trade-off between choosing an application-specific dynamics set has with models that do not have access to this knowledge in a different application. Moreover, one could add experiments where a misspecified model class is used to define the dynamics and show how the performance degrades wrt other algorithms that do not require this assumption.

In Section 5.1, the last experiment with distortion parameter,  the choice of distortion per simulator makes more sense than having an unlimited number of simulators to test robustness to distortion and hence makes me wonder (again) about how this model would perform against other model-based/adaptive algorithms.

On a minor note, the authors should make the setup in section 3 a little more self-sufficient and motivate the problem from a recommender system perspective i.e. how stochasticity and non-stationarity come into play.

Overall, I find the paper convincing and novel and would recommend acceptance.

---

> ### Author Response · Authors · 2020-11-20
> **Reply to AnonReviewer3**
>
> We thank the reviewer for the constructive suggestions and the acknowledgment of the practical value of our work. We address the detailed concerns in the following.
>
> A: what is the trade-off between choosing an application-specific dynamics set has with models that do not have access to this knowledge?
>
> The problem is worth discussing. In this work, we trade-off the dynamics with human knowledge and without human knowledge via the mapping function mentioned in section 4.3. In particular, in our application, for the dynamics hard to design by rules (e.g., the completed orders of a driver for each day), we learn to predict via a neural network. After that, we compute the other features of the next states by hand-coding rules (e.g., computing the cost via the predicted completed orders and platform recommendation actions). We think it is an effective and general solution for real-world applications: constructing application-specific rules that have access to the knowledge and inferring the left of features by learning. It makes use of expert knowledge and the ability of learning. More details can be seen in the section "The details from training to the deployment process" of "Reply to Common questions".
>
> B: A little more grounding to a sequential recommender system setting. e.g., how stochasticity and non-stationarity come into play? Motivating the distinction between other model-based and other adaptive policy learning methods
>
>
> Thanks for the useful advice. We have update the additional background and related work from the perspective of SRS. Please check the revision paper for more details.
>
> C: how this model would perform against other model-based/adaptive algorithms.
>
> The experiments “LTS3” in Figure 2(c) are conducted with the same setting with “beta=0” with limited simulators in Figure 3(a). It can be seen that although the performance declines when the distortion level becomes larger, the performance is at least similar to the compared methods. In particular, for beta<=4, the performance is better than all other algorithms with beta=0. When 6<=beta<=8, the performance is similar to DR-OSI and still better than other algorithms.
>
> D: one could add experiments where a misspecified model class is used to define the dynamics and show how the performance degrades wrt other algorithms that do not require this assumption.
>
> The method “DIRECT”  train in the dynamics model where environment parameters are nearest to the target domain and deploy direct.   It can be regarded as the experiments where a misspecified model class is used to define the dynamics.  Results in Figure (2) show that the performance degradation is severe in the tasks.
>
> We also add another model-based method DEMER-policy to the semi-online test. DEMER-policy train policy in the environment with the same domain to the target environment, but the dataset is collected in another time periods. The results shows the performance in unstable in different cities. We surmise that the performance is depended on the difference of the environments.

---

### Official Review · AnonReviewer2 · 2020-11-03
**Paper 652 Comments**

**Rating:** 7
**Confidence:** 3

**Review:**

This paper discusses the problem of RL based sequential recommendation system and proposes a model-based learning method to solve the offline RL in real-world applications. It addresses the two challenging problems:  (i) real-world recommendation
environments are usually non-stationary, and (ii) real-world environments recommendation are often with stochasticity. The proposed model tackles these two problems by learning to adapt to different simulators generated by the offline dataset. Experiments demonstrate the effectiveness of the proposed model against state-of-the-art baselines on one dataset. This paper could be improved in the following aspects:

1. Implementation code is not released.
2. More details of the deployment of the model should be provided.
3. Only a few baseline methods are introduced with the proposed model. More RL based recommendation models and simulator models should be incorporated to demonstrate the effectiveness of the model.

---

> ### Author Response · Authors · 2020-11-20
> **Replay to AnonReviewer2**
>
> We thank the reviewer for the acknowledgment of our contribution and for the constructive comments.
>
> Response to the detailed comments:
> 1. We are refactoring our project now. We will add the Github link in the camera-ready version and the reproducible code will be released on Github.
> 2. Thank you for the suggestion. There are several interesting implementation details from training to the deployment process. It could be helpful to inspire other researchers. Please refer to the section "The details from training to the deployment process” of "Reply to Common questions"  for more details. We will update the deployment details in appendix F.
> 3. We have updated another baseline method called "DEMER-policy" which is a model-based RL based recommendation models in the semi-online test. Please refer to the section "New baseline Algorithm” of "Reply to Common questions" for more details.

---

### Author Response · Authors · 2020-11-20
**Reply to common questions**

### The details from training to the deployment process

(1) How to construct a mapping function?

Given a specific SRS application, we can split the state space into the following parts:
- Customer-related static states: the states which are invariant to the recommended actions and are fixed for each customer. For example, the age and gender of each driver. Since we model the process of interaction between the platform and a customer to an MDP,  the customer is fixed in a trajectory and thus customer-related states are invariant;
- Timestep-related static states: the states which are invariant to the recommended actions and are fixed for each time-step. For example, the weather for each day. We train and the dynamics model with the same time period as the collected dataset and thus the timestep-related states can be found from the dataset and fixed.
- Dynamics-related states: the states which are dependent on the given recommendation actions and hard to hand-code by rules. For example, the completed orders and GMV of a driver in a day given the predicted completed orders and the recommendation action.
- Rule-related states: the states which can be computed by rules given other states. For example, we can compute the cost of a driver in a day, given the completed orders and the recommendation action, and we can compute the averaged GMV of a driver by averaging the predicted completed orders of recent days. Instead of learning to predict all the states, the neural network is used to predict the "dynamics states".

After sampling an initial state $s_0$ from the dataset at the beginning, then the trajectories of the environment model can be sampled by the following steps:
1. fix the customer-related static states: $s_t^{customer} = s_{t-1}^{customer}$;
2.  set the timestep-related static states $s_t^{timestep}$ with  the state in dataset ;
3.  predict the dynamics states by neural network model: $s_t^{dynamics} = NN(s_{t-1}, a_{t-1})$, where $NN$ denote the trained neural network;
4.  update the constructable states by rule: $s_t^{rule} = mapping(s_{t-1}, a_{t-1}, s_t^{timestep}, s_t^{dynamics}$);
5.  put it together: $s_t = [s_t^{customer}, s_t^{timestep}, s_t^{dynamics}, s_t^{rule}]$.

The framework of semi-dynamics model learning is general to other problems since it is easy for human experts to construct some rules for some states with a well-defined relationship and any number of rules can be embedded to reduce the complexity of learning. The more rules we have, the less learned states are needed. In our application, we have 54 dimensions of customer-related static states, 6 dimensions of timestep-related static states, 3 dimensions of dynamics states, 140 dimensions of rule states.
#### (2) how to partition the offline data to exhaust possible environment models?
The problem is indeed harder to give a general solution without knowledge of the environment. But in SRS, there are several common choices: First is splitting by time periods since the customer behaves differently across time periods [1]. The second is splitting by some identification category features in the application since it often represents the domain classes. In our application, we split the dataset by the month and city id.
#### (3) how to estimate the performance of the learned environment models?
We split the dataset into a training set (80%) and a testing set (20%), the environment model is learned in the training set and evaluated in the testing set. We design multiple metrics that are concerned by the operational staff. For example, the similarity of total completed orders of drivers in a city between the data run by the environment model and the corresponding real-world data.   We use Pareto improvement (a new model where some metrics will gain, and no metrics will lose) of the metrics in the testing set to define the performance improvement of the environment model.
### New baseline algorithm
We have updated another baseline method called "DEMER-policy" in the semi-online test [2].  It is a model-based RL based recommendation models which remedy the extrapolation error of offline training by selecting the offline dataset nearest to the deployment environments (via human knowledge) and learning model with generative adversarial imitation for better generalization ability. In particular, it trains a dynamics model for each domain (e.g., city) and learns policies for each dynamics model. When deployment, it deploys the policies in the domain where it trained. The dynamics model and policy are retrained periodically.

### Reference
[1] Philip S. Thomas, Georgios Theocharous, Mohammad Ghavamzadeh, Ishan Durugkar, and Emma Brunskill.   Predictive off-policy policy evaluation for nonstationary decision problems,  with applications to digital marketing.

[2] Wenjie Shang, Yang Yu, Qingyang Li, Zhiwei Qin, Yiping Meng, and Jieping Ye. Environment reconstruction with hidden confounders for reinforcement learning based recommendation.

---

### Author Response · Authors · 2020-11-23
**Revision update**

We would like to thank the reviewer for helpful suggestions. We are glad that reviewers generally appreciated the contributions. We have updated our paper based on the suggestions. The revision mainly in the introduction, related work, background, method, experiment, and appendix F. We mark all modifications with blue color. The revision includes the following changes:
1. The preciseness and self-sufficiency of the description (in the introduction, related work, background, and method);
2. A new baseline algorithm (in the experiment)
3. The details of deployment (in appendix F)

We hope the revision could improve the clarity of the paper and its contribution and could improve the score of the work.  Thank you again for your patience in the review.

Best,

Authors

---

### Decision · Program_Chairs · 2021-01-07
**Final Decision**

**Decision:**

Reject

**Comment:**

This paper is rejected.

The authors focus on offline RL for the sequential recommender system problem and propose an approach that:
* builds multiple models based on splits of the offline data using domain knowledge
* splits the policy into a context extraction system and context conditioned policy (similar to Rakelley et al.)

While R1 and R4 appreciate the changes, they both feel that the paper is not ready for publication at this time. R1's main concerns is the generalizability of the proposed solution because it relies heavily on manually defined rules and domain expert knowledge. R4 was concerned with the definition and precision of robustness. How is robustness quantified? Finally, many of the baselines were not built for partially observed environments, so it is unsurprising that they perform poorly. Baselines with recurrent policies would strengthen the paper.